# Olfactory receptor neurons generate multiple response motifs, increasing coding space dimensionality

Brian Kim[1,2†], Seth Haney[3†], Ana P Milan[4], Shruti Joshi[3], Zane Aldworth[1], Nikolai Rulkov[5], Alexander T Kim[1], Maxim Bazhenov[3*], Mark A Stopfer[1*]

[1]Eunice Kennedy Shriver National Institute of Child Health and Human Development (NICHD), National Institutes of Health (NIH), Bethesda, United States; [2]Brown University - National Institutes of Health Graduate Partnership Program, Providence, United States; [3]Department of Medicine, University of California, San Diego, San Diego, United States; [4]Department of Clinical Neurophysiology and MEG Center, Amsterdam Neuroscience, Vrije Universiteit Amsterdam, Amsterdam, Netherlands; [5]Biocircuits Institute, University of California, San Diego, La Jolla, United States

**Abstract** Odorants binding to olfactory receptor neurons (ORNs) trigger bursts of action potentials, providing the brain with its only experience of the olfactory environment. Our recordings made in vivo from locust ORNs showed that odor-elicited firing patterns comprise four distinct response motifs, each defined by a reliable temporal profile. Different odorants could elicit different response motifs from a given ORN, a property we term motif switching. Further, each motif undergoes its own form of sensory adaptation when activated by repeated plume-like odor pulses. A computational model constrained by our recordings revealed that organizing responses into multiple motifs provides substantial benefits for classifying odors and processing complex odor plumes: each motif contributes uniquely to encode the plume's composition and structure. Multiple motifs and motif switching further improve odor classification by expanding coding dimensionality. Our model demonstrated that these response features could provide benefits for olfactory navigation, including determining the distance to an odor source.

*For correspondence:
mbazhenov@health.ucsd.edu
(MB);
stopferm@mail.nih.gov (MAS)

†These authors contributed
equally to this work

Competing interest: The authors
declare that no competing
interests exist.

Reviewing Editor: Piali
Sengupta, Brandeis University,
United States

## Editor's evaluation

This important work describes the temporal mechanisms of odor coding in the olfactory neurons of the locust. The supporting evidence is compelling and based on extensive experimental and computational analyses. This work will be of interest to sensory neuroscientists.

## Introduction

Odors provide many types of important information about the environment, and are characterized by their chemical compositions and concentrations. Unlike vision or audition, which, in most basic form, can be described with the two dimensions of intensity and frequency, the tens or hundreds of thousands of detectable odorant molecules come in many different shapes, sizes, and charge distributions, an assortment of attributes requiring a high-dimensional description. Also, odorants often reach detectors as chaotic and turbulent plumes comprised of odorized pulses separated by expanses of clean media, air or water. The structures of odor pulses within a plume are shaped by many factors, including the medium's speed, environmental features such as hills, trees, or buildings, and the detector's distance from the source. Therefore, the timing of odorants reaching a

sensor can convey important information about the surroundings. To make use of this information, olfactory systems must generate high-dimensional representations of both chemical and timing information in a format usable by downstream circuits. This task is challenging, yet animals rely on the results for survival. Notably, olfaction is achieved by relatively few layers of neurons. Further, similar anatomical structures and physiological mechanisms for processing odors appear in widely divergent species, suggesting evolution has converged upon an optimized set of solutions to olfactory challenges.

In animals, the frontline encoders of odors are arrays of olfactory receptor neurons (ORNs), each expressing one of many types of olfactory receptor protein. The number of receptor types varies with species, from 119 in the desert locust (*Pregitzer et al., 2017*) to ~400 in humans (*Malnic et al., 2004*; *Trimmer et al., 2019*), and over 1000 in mice (*Zhang and Firestein, 2002*). Individual ORNs may be tuned narrowly or broadly , and typically, many types of ORNs respond to any given odor (*Hallem and Carlson, 2006*). Together, their responses give rise to high-dimensional, combinatorial representations that encode information about the environment. Because ORNs provide all the information available to the animal about the olfactory environment, it is important to understand the diversity and complexity of their responses.

To better characterize ORN response dynamics, we made extracellular recordings from the locust antenna while presenting odor pulses individually, or in regular trains, or as realistic, chaotic odor plumes. We found that ORNs can generate four distinct types of firing pattern motifs. While ORNs respond reliably with a given motif to a given odor across multiple trials, we also found that a single ORN can respond to different odors with different motifs, a phenomenon we term 'motif switching.' Further, when elicited by repeated odor pulses, the four response motifs undergo distinct forms of adaptation. With computational modeling, we found these novel aspects of stimulus encoding contribute substantially to classifying odors and to processing complex natural odor plumes, including extracting odor-invariant information about distance to the odor's source. Together, these results reveal new ways ORNs utilize the temporal domain to expand their coding space dimensionality.

## Results
### ORNs can respond to odors with four distinct firing patterns

To characterize the response properties of ORNs, we made extracellular recordings from antennae of intact locusts by placing electrodes against the base of sensilla while delivering pulses of odors well-separated from each other in time (40 animals, 62 sensilla, 198 odor-ORN pairs) (*Figure 1A*). Because sensilla contain multiple ORNs (*Ochieng et al., 1998*), we then analyzed the recorded waveforms with a spike sorting algorithm to assign odor-elicited spikes to individual ORNs (*Figure 1—figure supplement 1*). We recorded mainly from trichoid sensilla because they contain small numbers of ORNs (3–5, *Ochieng et al., 1998*) making spike sorting tractable, but we found unsorted, population activity recorded from other types of sensilla yielded results consistent with the responses of sorted ORNs, including prominent onset and offset activity (*Figure 1—figure supplement 2*). All results other than those shown in *Figure 1—figure supplement 2* are based on recordings from trichoid sensilla.

We used an unsupervised hierarchical clustering method to group odor-elicited responses, pooled across odor-ORN combinations, into categories (*Figure 1B*). Notably, we found the responses, particularly to longer stimuli, did not fall into a continuum of patterns, but rather clustered into distinct motifs; a boot-strap analysis indicated that four motifs provided the best clustering of responses (see 'Materials and methods,' *Figure 1—figure supplement 3*). We termed these four response motifs excitatory, delayed, offset, and inhibitory (*Figure 1C*, *Figure 1—figure supplement 4*; see 'Materials and methods'). Excitatory responses featured a sharp increase in firing rate at the immediate onset of the odor and decayed rapidly back to baseline even as the stimulus persisted. Delayed responses featured a slower increase in firing rate and a gradual decay back to baseline throughout an odor pulse. Inhibitory responses featured a sharp decrease in firing rate during an odor presentation that returned to the baseline level after the odor was removed. Offset responses were also inhibited during odor presentation but, upon the removal of odor, immediately showed an increase in firing rate exceeding the baseline level. Also, 1 s and 4 s odor pulses elicited similar results (*Figure 1C*).

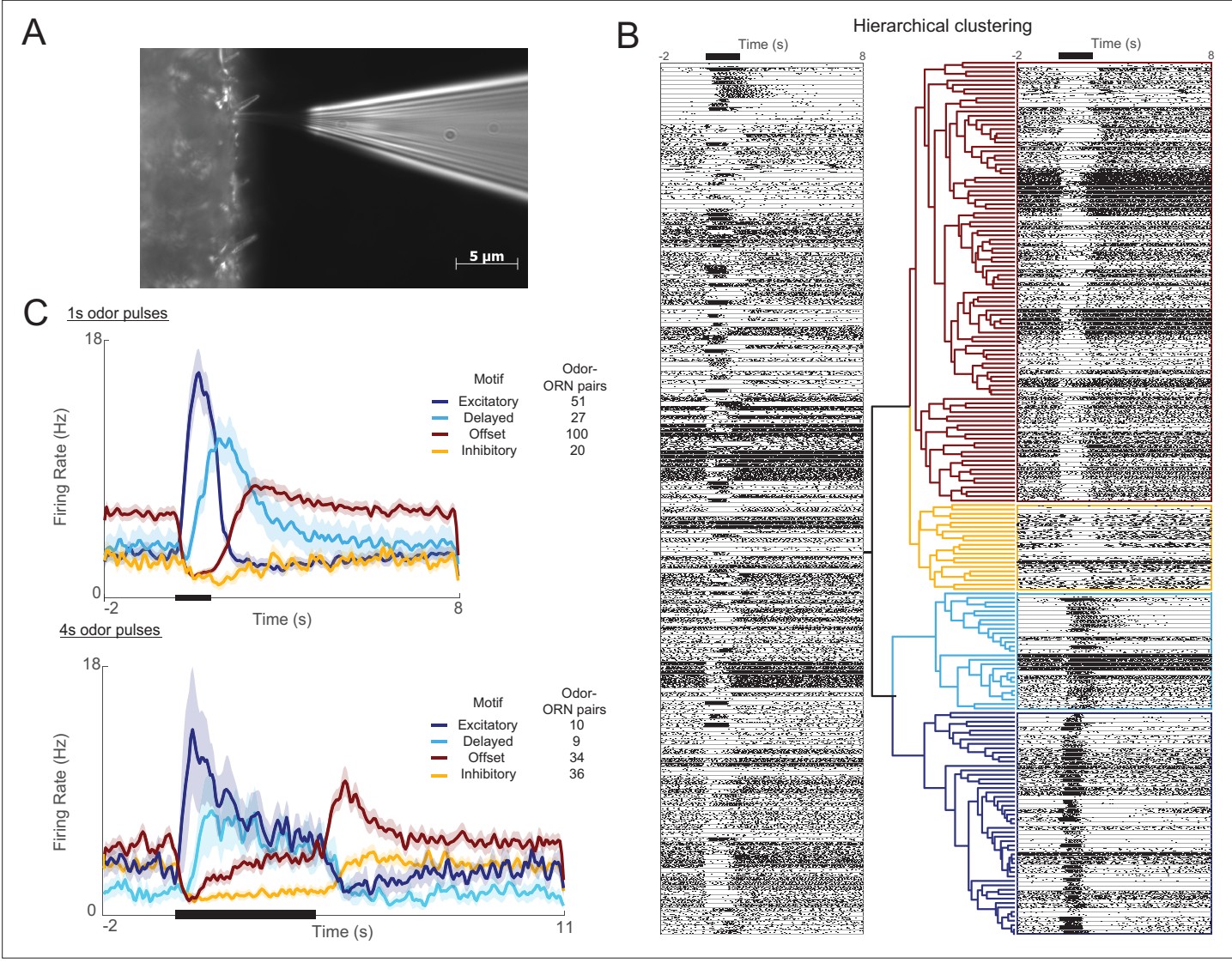

**Figure 1.** Olfactory receptor neuron (ORN) responses cluster into four distinct motifs. (**A**) Extracellular recordings were made with a blunt quartz electrode placed on the base of the sensillum. (**B**) Raster plot shows ORN spikes elicited by 1 s odor pulse (black bar at top). Horizontal lines separate ORN-odor pairs (five trials each). A hierarchical clustering algorithm (see 'Materials and methods') grouped these responses into four distinct motifs. (**C**) Histograms show the four motifs (excitatory, delayed, offset, and inhibitory) reliably elicited by 1 s and 4 s odor pulses. Five trials each; bold lines: means; shading: SEM.

The online version of this article includes the following figure supplement(s) for figure 1:

**Figure supplement 1.** Example of a sensillum recording including two olfactory receptor neurons (ORNs).

**Figure supplement 2.** Basiconic sensilla, which contain many olfactory receptor neurons (ORNs), show response patterns consistent with motifs observed in trichoid sensilla.

**Figure supplement 3.** Statistical significance of response motifs found through hierarchical clustering.

**Figure supplement 4.** Odor-olfactory receptor neuron (Odor-ORN) responses to 1 s odor pulses, organized by response motif.

## A given ORN can generate more than one response motif

Notably, a given ORN could respond with different patterns of spiking when presented with different odors. For example, from the same ORN, a brief pulse of pentyl acetate reliably elicited a delayed response motif, but an identical pulse of cyclohexanol reliably elicited an excitatory response motif (*Figure 2A*). This phenomenon, termed response motif switching, was not rare, occurring with probabilities ranging from 0.31 to 0.66 (mean = 0.38) for different pairs of odors (see *Figure 2B–D*). By contrast, we found very little switching across repeated trials of the same odor (0.096), and an

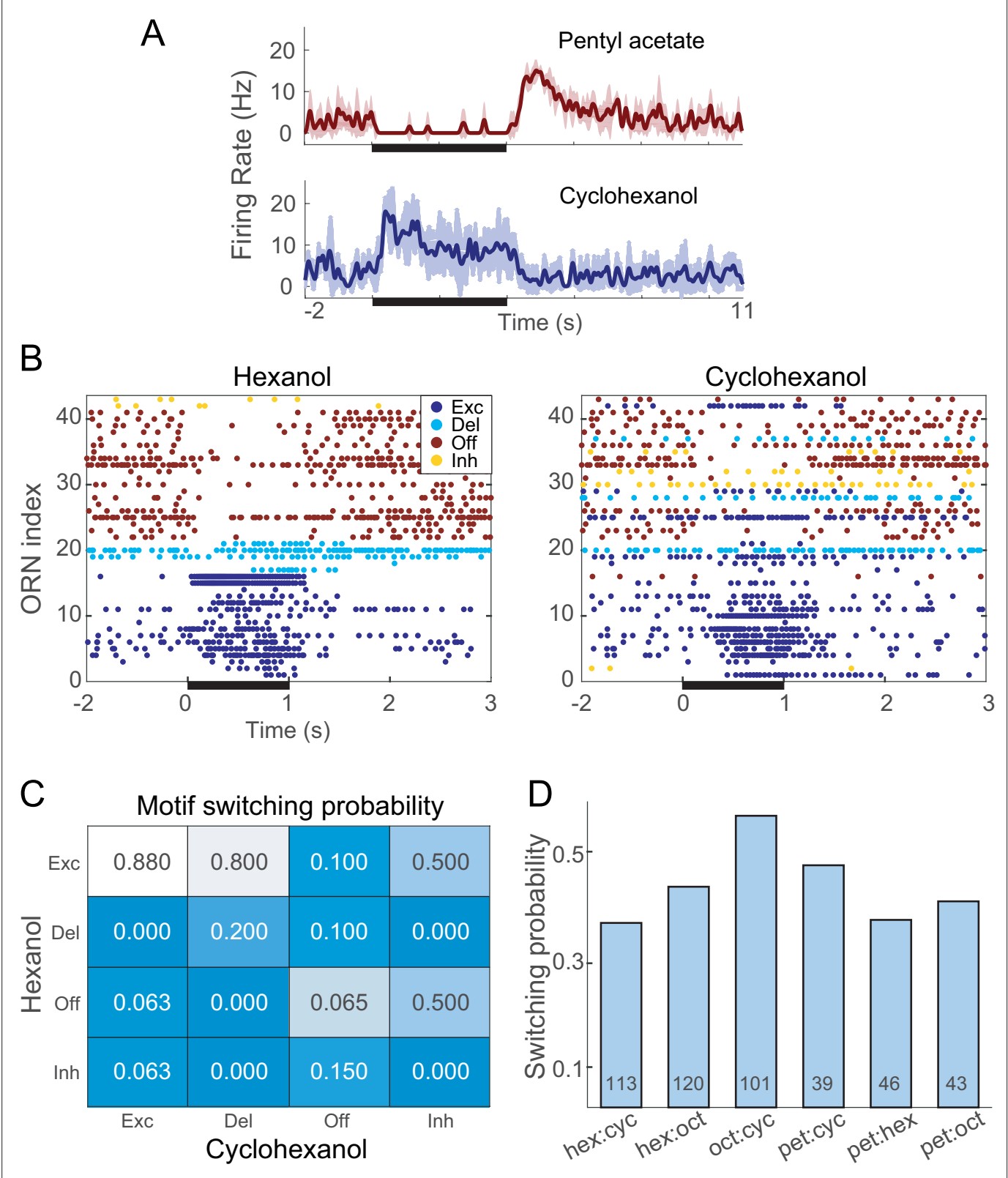

**Figure 2.** Different odors can elicit different response motifs from a given olfactory receptor neuron (ORN). (**A**) A single example ORN responds to two different odorants with different motifs. Five trials; bold lines: mean firing rate; shading: SEM. (**B**) Raster plots show spiking responses of 41 ORNs to 1 s (black bar) pulses of hexanol and cyclohexanol. Responses to both odors are sorted by motifs elicited by hexanol. (**C**) Conditional probability of an ORN

*Figure 2 continued on next page*

*Figure 2 continued*

producing a specific motif in response to hexanol given its response motif to octanol. (**D**) Switching probabilities varied with odor pairs (numbers at bottoms of bars: numbers of tested ORNs).

The online version of this article includes the following figure supplement(s) for figure 2:

**Figure supplement 1.** Response motifs vary little with odor concentration.

intermediate amount of switching across concentrations of the same odor (0.25, see *Figure 2—figure supplement 1*). Thus, motif switching appears to convey information about odorant identity and concentration in a way that is robust to trial-to-trial variation.

## Different ORN response motifs have different adaptation profiles

ORN responses adapt when odor pulses are lengthy or repeated rapidly (*Barrozo and Kaissling, 2002*; *Bau et al., 2002*; *Lemon and Getz, 1997*; *Marion-Poll and Tobin, 1992*). To characterize response adaptation properties of each ORN response motif, we delivered repeated 200 ms odor pulses at a range of inter-pulse intervals (IPIs) (see 'Materials and methods'; *Figure 3A*, *Figure 3—figure supplement 1*, *Table 1*). Notably, each response motif showed a distinct adaptation profile (*Figure 3B*; see 'Materials and methods' for details of analysis). The peak of excitatory motif responses decreased significantly over a train of odor pulses presented at brief, 0.5 s IPI. However, under the same stimulus conditions, the peaks of offset motif responses increased significantly, even with IPIs as long as 1.0 s; the average final response to an IPI of 0.5 s was threefold larger than the first. The delayed motif tended to increase after the first odor pulse, but this trend fell just short of statistical significance (*Table 1*). The inhibitory motif appeared to show no adaptation.

## Computational modeling of ORN response motifs

The response motifs we observed in vivo seemed likely to contribute to the processing of olfactory information. To test this idea, we designed a computational model based on observations made in vivo. Using the responses of ORNs to 4 s pulses of odor as templates, we constructed models of individual neurons to represent each response motif (see 'Materials and methods'; *Bazhenov et al., 2008*; *Rulkov, 2002*; *Rulkov et al., 2004*); this approach allowed us to model a biologically realistic population of 10,000 ORNs. The models included realistic levels of input noise and variations in baseline activity. Individual trials were generated by creating different random instantiations for the input noise (see 'Materials and methods'). The magnitude of each odor-specific ORN response was determined by the angle between two high-dimensional vectors: an ORN-specific chemical selectivity vector, $\mathbf{V_{ORN}}$, and an odor-specific characteristic vector, $\mathbf{V_{odor}}$ (see 'Materials and methods'), so similar odors elicited similar responses. As desired, model ORN responses to a single odor pulse (*Figure 4A and B*) and trains of pulses (*Figure 4C*) matched those found in vivo (see *Figures 1 and 3*). The model's responses to pulse trains were set to adapt in motif-specific fashions as found in vivo, with the excitatory motif responses decreasing and the offset motif responses increasing with each pulse (*Figure 4C*). As desired, our model provided an accurate simulation of ORN responses observed in vivo (*Figure 4—figure supplement 1* provides a quantitative comparison of response peaks and latencies in vivo and in the model).

We used this model to determine how the diversity of ORN motifs contributes to the complexity of the odor response. In principle, more complex odor responses allow for higher coding dimensionality. We quantified response complexity by performing principal component analysis (PCA) and evaluating the number of components needed to explain the variance across the entire population; a more complex odor response would require more principal components to explain the same amount of variance. We compared responses encoded only by excitatory motifs to responses including all motifs using numbers of ORNs matched between these two cases (N = 1374 each). When only excitatory motifs were included in the analysis, the first component alone explained nearly 30% of the variance and the second component added only ~1%. In contrast, when all motifs were included, the first component only explained ~15% of variance and 27 principal components were required to explain 30% of the variance (*Figure 4D*). Thus, the existence of four response motifs dramatically expanded the dimensionality of the odor representation.

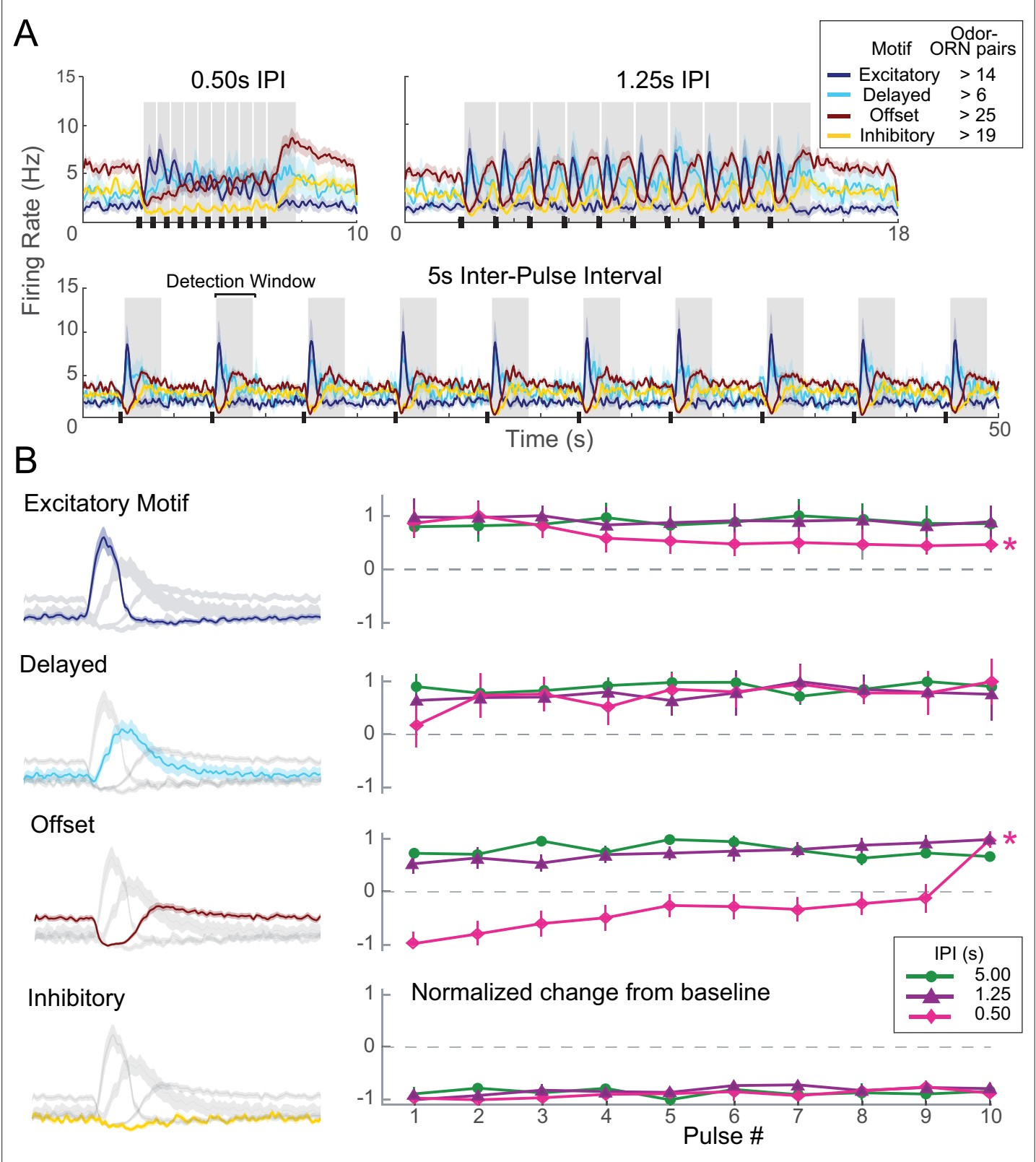

**Figure 3.** Each olfactory receptor neuron (ORN) response motif has a distinct adaptation profile. (**A**) Responses of ORNs, grouped by motif, to odors pulsed at different inter-pulse intervals (IPIs). Peaks for different motifs and pulses were measured as maximum absolute change from baseline within detection windows (shaded areas). (**B**) Adaptation characteristics of excitatory, delayed, offset, and inhibitory response motif to 10 pulses for each IPI. Left: response motifs; Right: normalized response change from baseline. *Statistically significant changes elicited by odor pulses delivered at 0.50 s

*Figure 3 continued on next page*

*Figure 3 continued*

IPI. Excitatory motif responses significantly decreased; offset motif responses significantly increased; delayed motif responses modestly increased; and inhibitory motif responses did not change. See *Table 1* for statistical tests.

The online version of this article includes the following figure supplement(s) for figure 3:

**Figure supplement 1.** Additional examples of motif-specific adaptation profiles.

## Response motif switching contributes substantially to odor classification

With our model, we could independently vary odor-elicited response motifs and response magnitudes (*Figure 4E*), allowing us to evaluate the extent to which motif switching benefited odor classification in a way that cannot be tested in vivo. Thus, we simulated a realistically large number of ORNs (10,000) and compared the relative success of classifying two different odors (odor 1 and odor 2) with three different versions of our model in which we systematically varied the probability of motif switching. Model version 1: the probability of switching response motif when switching from odor 1 to odor 2 was 0%; version 2: 10%; version 3: 50%. We found that the model versions that simulated higher motif switching probability made it easier to distinguish these two similar odors. *Figure 5A* shows the responses of 41 example ORNs. Trajectories of the responses of the 10,000 ORN population over time plotted in PCA-reduced space (see 'Materials and methods') increasingly separated as the probability of motif switching increased (*Figure 5B*), demonstrating that motif switching made the ORN population responses to the two odors more different from each other. When we independently varied the similarity of the odors and the probability of motif switching (see *Figure 4E*), support vector machine (SVM)-based classification of the ORN responses showed classification accuracy improved as motif switching probability increased for all degrees of odor similarity (*Figure 5C*). With 4 s odor pulses, even the lowest tested probability of motif switching (10%) substantially improved odor classification to the point that it made a difficult classification task (when test odors were very similar) as successful

**Table 1.** Results of ANOVA tests for sensory adaptation experiments shown in *Figure 3B* and *Figure 3—figure supplement 1*.

| Excitatory motif | | | | Delayed motif | | | |
|---|---|---|---|---|---|---|---|
| IPIs (s) | Df | F-value | p-value | IPIs (s) | Df | F-value | p-value |
| 0.50 | 1, 9 | 3.11 | 0.002 | 0.50 | 1, 9 | 1.98 | 0.052 |
| 0.75 | 1, 9 | 0.51 | 0.865 | 0.75 | 1, 9 | 0.88 | 0.549 |
| 1.00 | 1, 9 | 0.80 | 0.614 | 1.00 | 1, 9 | 0.38 | 0.938 |
| 1.25 | 1, 9 | 0.84 | 0.576 | 1.25 | 1, 9 | 0.73 | 0.675 |
| 1.75 | 1, 9 | 0.38 | 0.941 | 1.75 | 1, 9 | 0.32 | 0.966 |
| 5.00 | 1, 9 | 0.67 | 0.734 | 5.00 | 1, 9 | 1.41 | 0.210 |

| Offset motif | | | | Inhibitory motif | | | |
|---|---|---|---|---|---|---|---|
| IPIs (s) | Df | F-value | p-value | IPIs (s) | Df | F-value | p-value |
| 0.50 | 1, 9 | 33.06 | 1.484E-37 | 0.50 | 1, 9 | 0.78 | 0.634 |
| 0.75 | 1, 9 | 6.59 | 4.663E-08 | 0.75 | 1, 9 | 1.05 | 0.406 |
| 1.00 | 1, 9 | 4.29 | 4.650E-05 | 1.00 | 1, 9 | 1.46 | 0.162 |
| 1.25 | 1, 9 | 2.94 | 0.003 | 1.25 | 1, 9 | 1.21 | 0.293 |
| 1.75 | 1, 9 | 1.32 | 0.229 | 1.75 | 1, 9 | 0.66 | 0.747 |
| 5.00 | 1, 9 | 1.76 | 0.077 | 5.00 | 1, 9 | 1.57 | 0.127 |

IPI: inter-pulse interval.

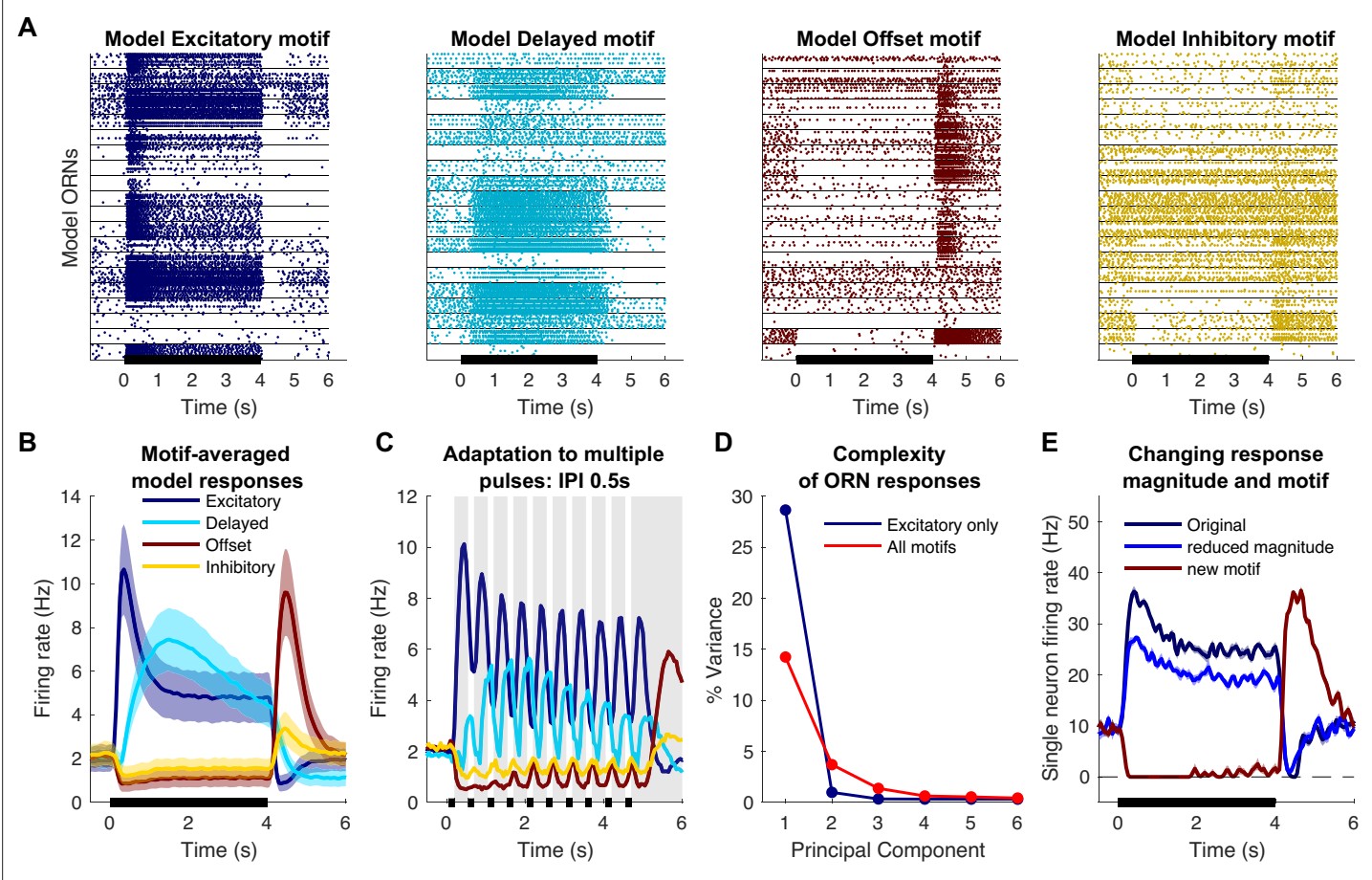

**Figure 4.** Computational model simulates response motifs. (**A**) Raster plots of simulated olfactory receptor neuron (ORN) responses illustrate each motif. Rows: different trials; horizontal lines separate responses of different ORNs. Black bar: odor pulse. (**B**) Firing rates averaged across trials and ORNs by motif. Bold line: mean; shading: SEM; compare to *Figure 1C*. (**C**) Model driven at 0.50 s inter-pulse interval (IPI) simulates adaptation profiles of each motif; firing rate averaged across ORNs within a motif; compare to *Figure 3A*. (**D**) Principal component analysis of excitatory only ORN responses (blue) or all response motifs (red) shows the inclusion of multiple motifs adds to the complexity of the responses. (**E**) The model can vary response magnitude and motif independently; responses of a single model ORN to three different odors are shown.

The online version of this article includes the following figure supplement(s) for figure 4:

**Figure supplement 1.** Distribution of peak firing rates and response latencies for different motifs in the model and in vivo elicited by a 1000 ms odor pulse.

as an easy classification task (when odors were very different). Together, these results demonstrate that motif switching, even if infrequent, can contribute substantially to successful odor classification.

## Response motifs represent complex odor plume features

The four response motifs of ORNs differed in their timing and adaptation properties, suggesting that they could contribute to encoding temporal features of olfactory stimuli. In natural environments, odorant molecules are usually arranged by a turbulent medium, air or water, into temporally complex plumes. We used the statistics of open-field measurements of real plumes (*Murlis et al., 2000*) to generate realistic, repeatable artificial plumes that included a distribution of 'burst' lengths (see 'Materials and methods'), and delivered them to the antennae of intact locusts while recording responses from ORNs in vivo (*Figure 6A*, top and middle). We also simulated the same plumes in our model (*Figure 6A*, bottom). The motif-dependent adaptation we had observed in vivo (*Figure 3A*) and reproduced in silico (*Figure 4C*) introduced motif-specific sensitivities to stimulus history. Notably, long bursts within a plume led to decreased excitatory motif responses but increased delayed motif responses (e.g., *Figure 6A*, ~9–12 s). On the other hand, short bursts led to increased excitatory motif

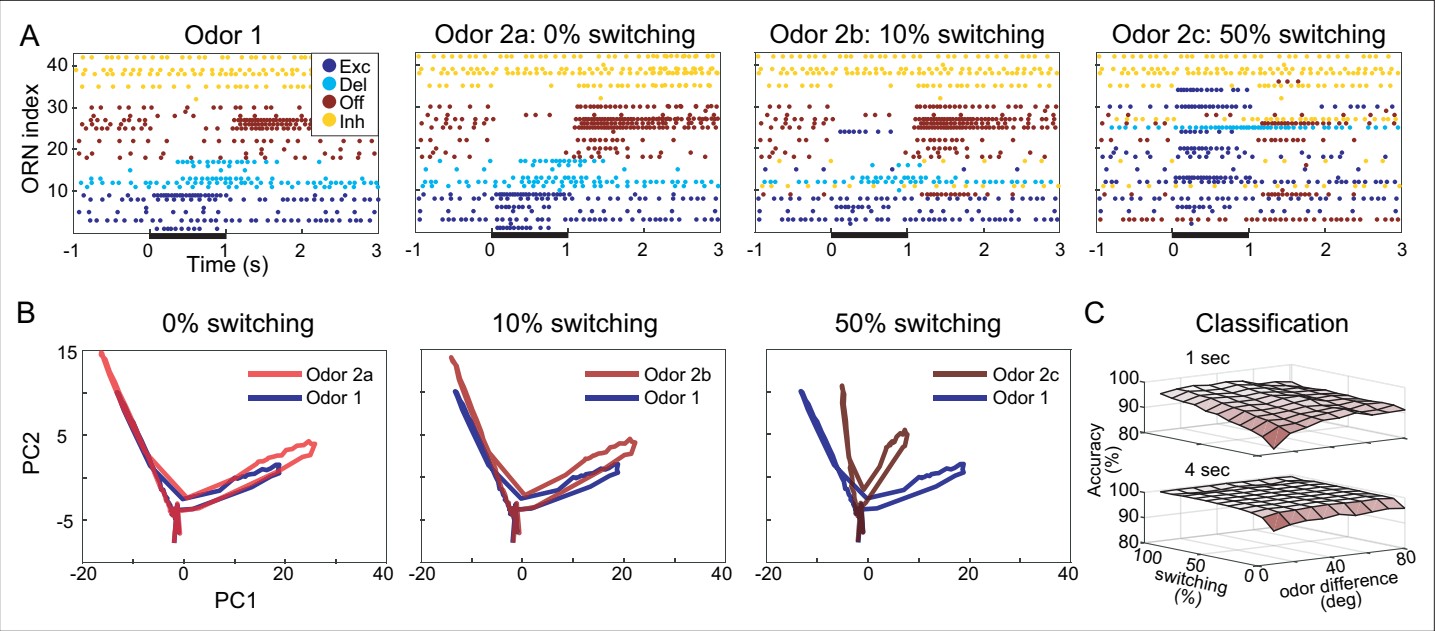

**Figure 5.** Computational model shows response motif switching substantially improves odor classification. (**A**) Simulated olfactory receptor neuron (ORN) spiking illustrates different motif switching probabilities. Odors 1 and 2 are similar (see 'Materials and methods'). Each ORN response is sorted by motifs elicited by odor 1. Raster plots show the responses to odor 2 become increasingly different from responses to odor 1 as motif switching probability increases. (**B**) ORN odor-elicited response trajectories in reduced principal component analysis (PCA) space show motif switching increases the separation between responses to similar Odors 1 and 2; response to odor 1 (blue) is the same in each panel; response to odor 2 (red) changes with switching probability. (**C**) Odor classification success as a function of odor similarity and motif switching probability for 1 s (top) and 4 s (bottom) stimulus pulses; even low switching probabilities improve classification performance; darker shading indicates lower classification accuracy. Odor similarity is quantified by angles (degrees) between odor vectors (see 'Materials and methods').

responses but did not change the delayed motif responses (e.g., *Figure 6A*, ~5 s). This motif-specific history dependence appeared likely to contribute to encoding temporally complex stimuli.

To quantify how each motif contributes to processing time-varying odor plumes, we applied a commonly used linear–nonlinear analysis scheme (*Butts et al., 2007*; *Geffen et al., 2009*). In this approach, the true firing rate of the neuron, $r(t)$, is predicted by convolving a linear filter, $f(t)$, with the time-varying stimulus, $s(t)$, and then the convolved product is thresholded with a nonlinear function, $g(x)$. This filter is useful because the waveform of the linear filter, $f(t)$, derived from this method precisely describes the sensitivity of the neuron to the history of the stimulus. To compute these filters, briefly, we deconvolved the stimulus, $s(t)$, from the firing rate data, $r(t)$, collected from ORNs. This approach allowed us to calculate filters for each ORN response in vivo and in the model (*Figure 6B* illustrates this approach; see 'Materials and methods'). As desired, these reconstructions accurately reproduced the firing rates of the ORNs from their trained filters on data not used for training.

As desired, filters generated from the model and from responses recorded in vivo matched closely (*Figure 6C*, *Figure 6—figure supplement 1*). We observed that different response motifs generated distinct filter waveforms, describing the different sensitivities of each motif to the temporal features of the odor plume. For example, filters for excitatory motifs showed that the ORNs generating them were most responsive ~0.4 s after odor filaments arrived (*Figure 6C*). To examine the stimulus-history dependence of the ORNs as a population, we embedded the filter waveforms of all the individual ORNs into an N-dimensional space (where N was the number of sample points for each filter) and used PCA to reduce each filter waveform into a single point in a two-dimensional space. We found that ORN filters clustered by motif in this space. Inhibition motifs (offset and inhibitory) and excitation motifs formed two ends of the principal filter axis, with the second dimension separating the delay motif from the other three (*Figure 6C and D*). This analysis revealed that each ORN response motif contributes uniquely to the olfactory system's representation of the odor plume's complex temporal structure.

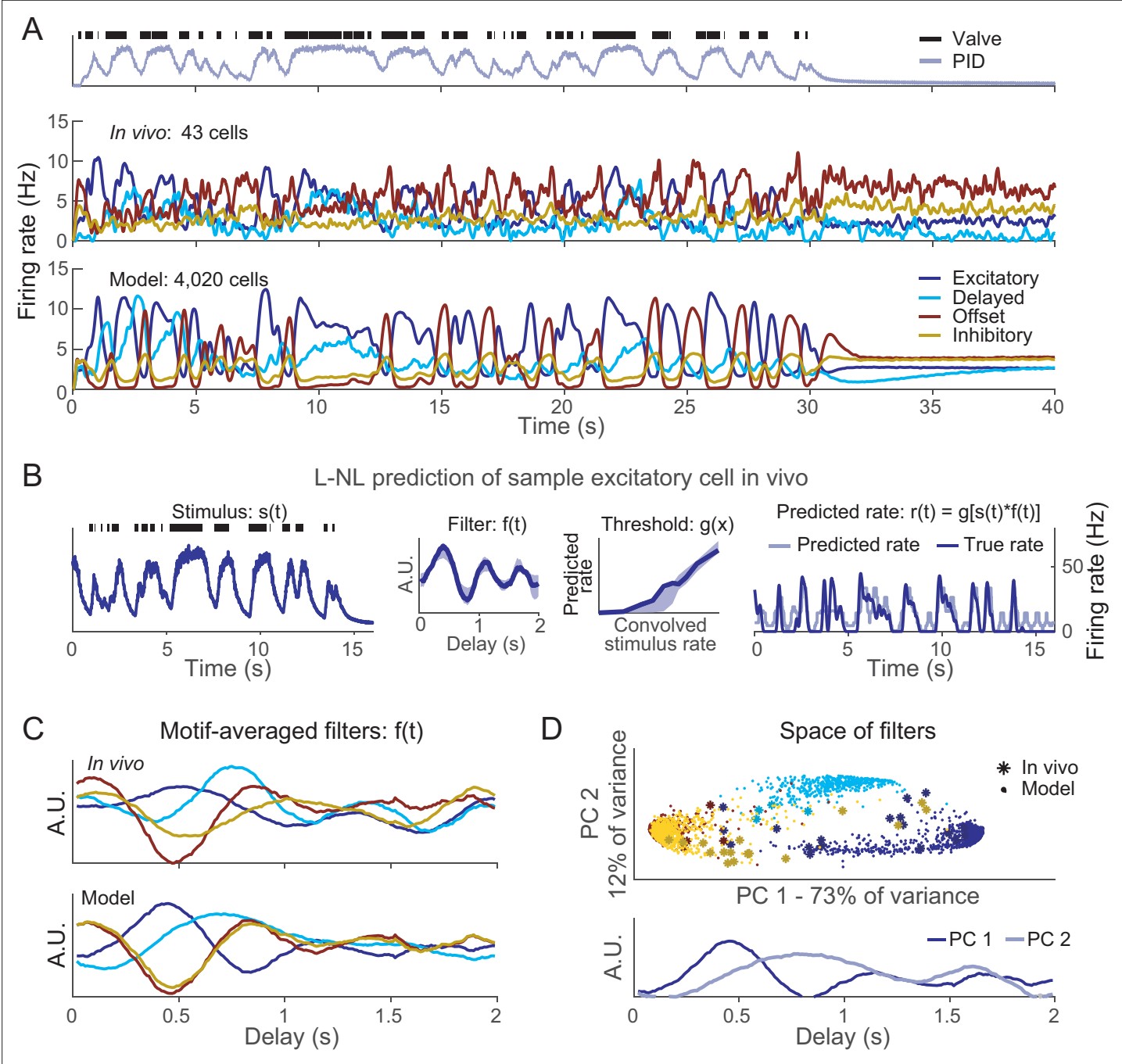

**Figure 6.** Responses to realistic odor plume stimulus. (**A**) Top: odor plume generated by opening and closing an olfactometer valve, measured by a photoionization detector (PID), and delivered to locust antennae in vivo. Plume represents sampling 2.5 m from odor source. Middle: responses of olfactory receptor neurons (ORNs) recorded in vivo; bottom: responses of model ORNs. (**B**) Linear–nonlinear prediction of excitatory-type ORN activity in vivo. The first 16 s of 40 s odor plumes were used for training and the next 16 s were used for testing. To assess the agreement, the test stimulus (left) was convolved with the learned filter and the result was passed to the learned threshold function (middle) to generate the predicted firing rate (right). Shading denotes a 95% confidence interval of prediction (see 'Materials and methods'). Stimulus correlations in naturalistic stimuli were accounted for (see 'Materials and methods'). (**C**) Averages of learned filters across all ORNs within a response motif in vivo (top) and in the model (bottom). Delay indicates time since the stimulus arrived at the antenna. (**D**) Principal component analysis of filters in the delay dimension. Filters of the same response motif cluster together. Top: filters colored by motif type. Bottom: two principal components in the analysis.

The online version of this article includes the following figure supplement(s) for figure 6:

**Figure supplement 1.** Statistical validation for distance-based classification with different motifs as shown in *Figure 6C*.

## Response motif diversity increases sensitivity to the distance to odor source

Our finding that different response motifs are sensitive to the different temporal features of a complex natural stimulus raised the possibility that this information could be used for ecologically relevant tasks that rely on assessing the timing of a stimulus. Published reports indicate a reliable relationship between the frequency of filaments within an odor plume and the distance to its source (*Murlis et al., 2000*). Further, evidence from walking fruit flies suggests that they navigate toward an odor source in response to both the overall frequency and inter-pulse intervals of their encounters with the odor plume (*Demir et al., 2020*; *Álvarez-Salvado et al., 2018*; *Jayaram et al., 2022*). To test whether, in principle, the diversity of ORN response motifs could help an animal determine the distance to an odor source (a key aspect of navigation), we simulated odor plumes characteristic of an odorant located 2.5 m, 5 m, 10 m, or 20 m away (*Murlis et al., 2000*). We used our model to generate responses of ORNs to these distance-specific plumes (*Figure 7A*; see 'Materials and methods').

We trained SVM classifiers to discriminate among the distances based on responses of the model ORNs (*Figure 7B*). We constructed different classifiers using only subsets of ORNs that respond with specific motifs, testing all possible combinations to assess their separate and combined contributions to classification success. We found that classifiers that included the excitatory and delayed motifs were most successful in determining distance to the source (*Figure 7C*). Excitatory and delayed motif responses were less correlated with each other than the other two possible combinations (*Figure 7D*), indicating that the excitatory and delayed motif responses encoded different information about stimulus timing. This result was robust to the duration of the response used to calculate both correlation and classification (*Figure 7E*) with longer durations increasing classification success. Together, these results show the different response motifs each extract distinct information that could be used to navigate toward odor sources.

## Discussion

Spiking patterns of olfactory neurons have been shown to convey information about the chemical composition of odors. In insects, projection neurons (PNs) generate spiking patterns that change with the identity and concentration of the odor. These complex dynamics arise in part from network interactions among PNs and local inhibitory neurons within the antennal lobe driven by heterogeneities in the temporal structures of odor-elicited ORN responses (*Martelli and Fiala, 2019*; *Raman et al., 2010*). Deeper in the brain, Kenyon cells (KCs), driven by PNs, change the timing of their spiking responses when the eliciting odor changes (*Gupta and Stopfer, 2014*), and even subtle changes in the timing of KC spiking can alter the responses of their follower neurons (*Gupta and Stopfer, 2014*). Because these information-bearing spiking patterns throughout the brain originate with the combinatorial spiking of populations of ORNs, it is important to understand the properties of these peripheral responses.

In natural settings, ORNs typically encounter odor plumes characterized by complex temporal structure. Turbulent flowing media churn the headspaces above odor sources into chaotic plumes (*Jacob et al., 2017*; *Levakova et al., 2018*; *Murlis et al., 1992*), and active sampling movements such as sweeping antennae or sniffing generate additional temporal structure (*Huston et al., 2015*). Animals can make use of timing information within odor plumes. Mice, for example, can use the separate timings of multiple odors comingled in plumes to isolate their separate sources (*Ackels et al., 2021*), and moths and mosquitoes will fly upwind only when attractive odors are appropriately pulsed (*Baker et al., 1985*; *Geier et al., 1999*). Although patterned activity originating in the olfactory environment may appear to conflict with patterned activity generated by olfactory neural circuits, combinatorial codes within the locust antennal lobes and mushroom bodies have been shown to disambiguate the two, representing not only the chemical identity and concentration of odors, but also their delivery timing (*Brown et al., 2005*).

We found that ORNs respond to odors with a distinct set of four motifs that can include immediate and delayed periods of spiking and periods of inhibition. Our computational analysis further revealed how each response motif contributes to processing complex temporal stimuli such as odor plumes. Excitatory and offset motifs formed two ends of a single encoding dimension, and the delayed motif formed another dimension. Thus, each motif, with its own sensitivity to specific temporal features of a

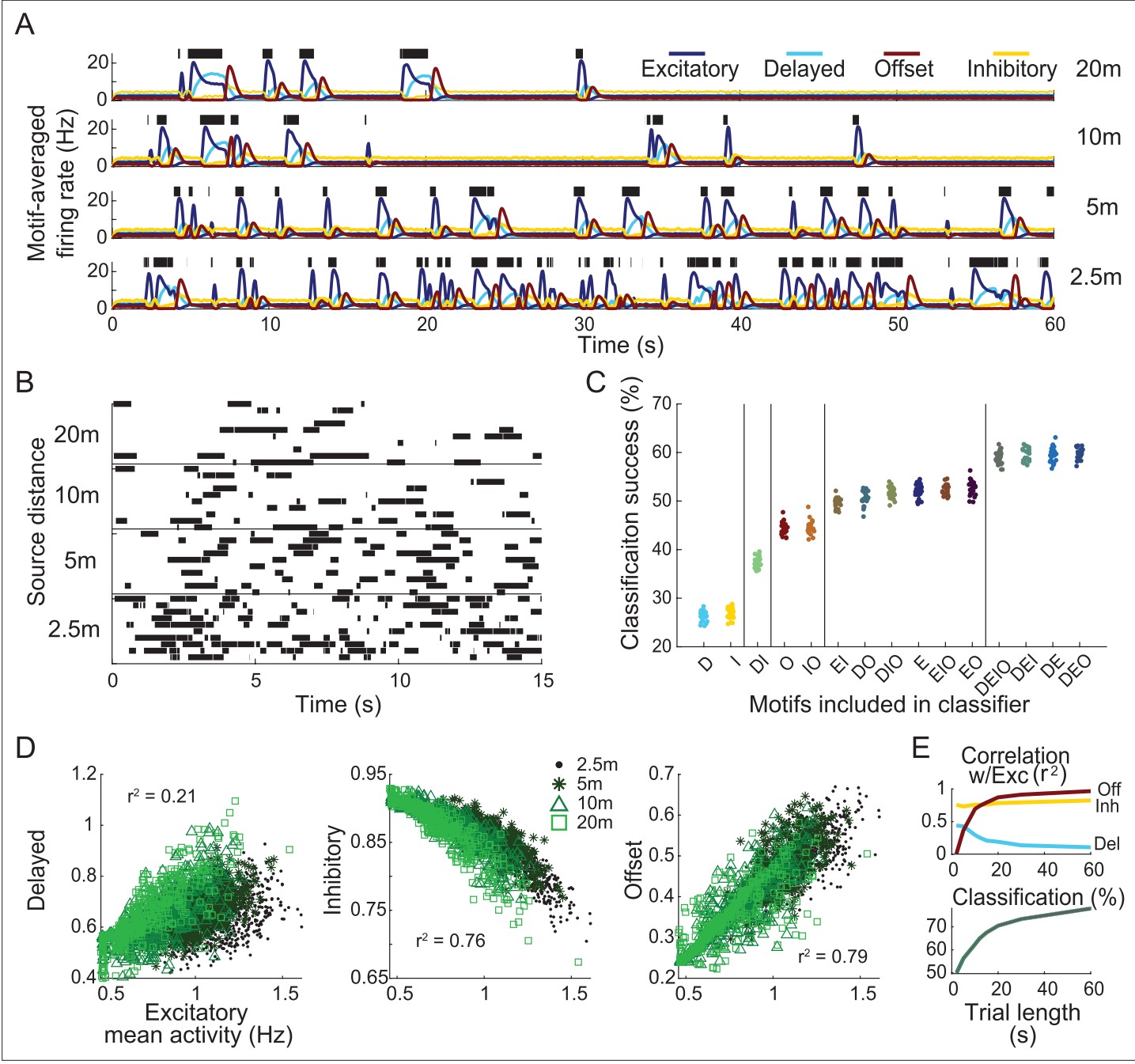

**Figure 7.** Response motifs encode information needed to determine distance to odor plume source. (**A**) Model olfactory receptor neuron (ORN) responses, averaged by motif, to four distance-dependent plumes (see 'Materials and methods'). Green bars: simulated olfactometer valve open times. (**B**) Depiction of stimulus classification task. Ten realizations of plume stimuli (trials) are shown for each distance. Plumes are colored by distance to source. The goal of the classification task is to correctly identify all plume stimuli of the same distance as belonging to the same group. (**C**) The success of distance-dependent plume classification. Each group of points represents 25 differently trained classifiers only including a subset of indicated response motifs. Vertical bars separate groups into statistically distinct sets, where any group within a set is statistically different from all groups outside this set (one-way ANOVA F(14,374) = 1830.2, p=0; see *Figure 3—figure supplement 1*). The x-axis labels denote which motifs are included in each analysis group: D, delayed; E, excitatory; I, inhibitory; O, offset. (**D**) Scatter plots show firing activity averaged over the entire 15 s window by motif. (**E**) Correlation between each motif and the excitatory motif calculated using different sample durations (top). Classification success for different sample durations using responses from all motifs (bottom).

stimulus, contributes in different ways to extracting timing information, and together provide a more thorough description of the stimulus to downstream neurons than that of any single motif.

Our experiments did not allow us to determine the molecular or neural mechanisms that generate each of the four response motifs. Earlier work established that the intrinsic dynamic properties of odorants, described as 'fast' or 'slow,' can contribute to variations in the timing of ORN responses (*Martelli et al., 2013*; *Su et al., 2011*). However, our experiments ruled out the possibility that intrinsic odorant dynamics underlie the response motifs we describe here. First, across our extensive dataset, all odors could elicit all four response motifs; second, photoionization detector recordings of our odor presentations all revealed 'fast' dynamics (not shown). It seems likely that 'slow' odors would elicit concentration-dependent elaborations in the response motifs we observed. In future work, it will be interesting to investigate the ways intrinsic odor dynamics interact with ORN response motifs. We predict such interactions would further increase ORN response dimensionality. Overall, our results suggest that these responses, together and in combination, play fundamental roles in olfactory processing.

We were surprised to find that a given ORN could generate different response motifs when stimulated by different odors. ORNs responded reliably and robustly with the same motif across nearly all trials of the same odor, but when the odor changed, the response motif sometimes changed. Motif switching was not rare; under some conditions in our sample, 66% of tested ORNs switched motifs when odors were switched. Even relatively subtle changes in odorant chemistry (e.g., hexanol to cyclohexanol), or changes in the concentration of an odor, could evoke changes in the response motifs of ORNs (e.g., immediate to delayed spiking), leading to dramatic differences in the timing of the odor response. Our computational model revealed that even infrequent motif switching improves odor classification by increasing the distance between the spatiotemporal representations of odors. The improvement was substantial, comparable to that obtained by changing test odor pairs from the most similar to the most different. Our study demonstrates a new property of ORNs: they can signal changes in an odor stimulus through discrete step changes in response motif as well as through continuous shifts in response magnitude. Motif switching thus appears to increase contrast between odors. Our experiments did not allow us to test whether all ORNs are capable of motif switching, or what, if any, organizational principle determines when a given ORN will switch motifs. But because some odor pairs were more likely than others to elicit motif switching, we speculate that motif switching may follow a logic that evolved to help discriminate ecologically relevant but chemically similar odors (*Su et al., 2012*). For these studies, we used odorants known to be ecologically relevant to locusts, including several found in the head space of wheat grass. Future experiments with larger sets of odorants, including blends or locust pheromones like 4-vinylanisole (4VA) and phenylacetonitrile (PAN), may help clarify the logic of motif switching.

Our work did not address the neural mechanism underlying motif switching. Odorant receptor proteins in ORNs may, for example, couple with multiple transduction pathways. Ephaptic coupling between ORNs cohabiting a sensillum could, in principle, allow one spiking ORN to change the firing pattern of its neighbor (*Su et al., 2012*; *Zhang et al., 2019*) in an odor-specific fashion, but our sensillar recordings did not reveal the anticorrelated spiking in pairs of ORNs expected of such an interaction (see *Figure 1—figure supplement 1*). Synaptic connections between ORNs have been identified in a connectomics analysis of the fly brain (*Schlegel et al., 2021*), and similar connectivity has been suggested to exist among gustatory receptor neurons (*Chu et al., 2014*). Such connectivity could potentially enable network interactions to generate multiple and switching response motifs, even at the sensory periphery.

Repeated stimuli elicit sensory adaptation, a form of short-term plasticity that encodes recent stimulus history. In agreement with prior work (*Barrozo and Kaissling, 2002*; *Bau et al., 2002*; *Lemon and Getz, 1997*; *Marion-Poll and Tobin, 1992*), we found ORNs adapted when they were activated by repeated pulses of an odor. However, we also found, surprisingly, that different response motifs adapted in distinct ways, including decreasing or increasing response intensity across a series of odor presentations. This response diversity was undetectable by previous studies employing electroantennogram (EAG) recordings, which sum the activity of populations of ORNs (*Huston et al., 2015*). The individual odor filaments comprising natural odor plumes vary from brief to long, each evoking, respectively, lesser or greater amounts of sensory adaptation in ORNs. Thus, adaptation adds contrast to the olfactory system's description of the plume's temporal structure. Additionally, each response

motif adapts differently, enriching the description by highlighting different time-varying aspects of the plume. To analyze these responses, we used our recordings made in vivo to generate filters that provide optimal descriptions of each ORN's sensitivity to changes in the stimulus with respect to time, including motif-specific responses and adaptation. These results show that different response motifs highlight different temporal features of a stimulus and, together, the combination of distinct response motifs with different forms of adaptation improves representations of temporal features in complex stimuli, including plume onset, offset, and the spacing between odor filaments. While our analysis did not investigate the mechanisms underlying the different forms of adaptation we observed, recent studies of ORNs in *Drosophila* reveal calcium dynamics play a role in adaptation (*Martelli et al., 2013*; *Martelli and Fiala, 2019*).

We examined whether timing information available in the multiple and switching response motifs in ORNs could provide practical benefits to animals. It has been shown that the structure of natural odor plumes contains information about the environment. For example, the length and spacing of odor filaments within plumes vary reliably with the distance to the odor source (*Murlis et al., 2000*). Further, behavioral (*Demir et al., 2020*; *Álvarez-Salvado et al., 2018*) and modeling studies (*Jayaram et al., 2022*) have shown that the first layer of the insect olfactory system extracts information sufficient for navigation from the temporal statistics of plumes. Using our model, we evaluated how the multiple response motifs generated by the ORNs could contribute to extract this information. We classified responses of the ORN population to odor plumes designed to simulate different distances to the odor source. We found that the distance to the odor source could be determined readily from information contained within the ORN responses. Notably, providing multiple ORN response motifs significantly improved the ability to classify source distance. Our model predicted that combining excitatory and delayed motifs provided the most successful way to determine distance to the source, suggesting that these motifs extract complimentary information about the odor statistics most needed to characterize distance. Other motifs likely contribute to encoding different odor features. Earlier work performed in vivo and with computational models has investigated the ways insect olfactory systems encode distance-dependent plume statistics (e.g., *Jacob et al., 2017*; *Levakova et al., 2018*). A recent modeling study found that adaptation of ORN responses contributes substantially to the sparsening of responses downstream, and to the sampling of the statistics of odor encounters that could aid navigation to food sources (*Rapp and Nawrot, 2020*). A common theme of these findings is that olfactory systems include mechanisms to extract information needed for navigation.

Earlier work has shown that odor-elicited responses of ORNs can be more complex than simple bursts of spikes that track the stimulus. For example, locust ORNs were shown to generate heterogeneous responses (*Raman et al., 2010*) and, in *Drosophila*, ORNs have been shown to produce either excitatory or inhibitory responses to odorants (*de Bruyne et al., 2001*; *Su et al., 2012*). Similar responses have also been observed in the ORNs of aquatic organisms, including lobster (*Bobkov et al., 2012*), where a subpopulation of lobster ORNs has been shown to rhythmically burst in a fashion essential for determining the edges of a turbulent plume during odor tracking (*Michaelis et al., 2020*). Recent results in *Caenorhabditis elegans* show that hedonic changes in olfactory context can change the responses of an ORN (*Khan et al., 2021*). However, to our knowledge, our results provide the first evidence that ORNs can generate a well-defined set of discrete temporally structured response motifs for different odors.

ORNs provide the olfactory system's first of several stages of signal processing. In insects, ORNs pass information to the antennal lobe, where spiking patterns originating in the periphery drive further processing in networks of local and projection neurons, leading to richer, higher dimensional olfactory representations (*Geffen et al., 2009*; *Raman et al., 2010*). Our study did not address how multiple and switching response motifs in ORNs affect the responses of downstream neurons such as LNs and PNs. Our earlier work established that antennal lobe circuitry generates high-dimensional, complex temporally structured responses, but only when it is driven by input with heterogeneous timing from ORNs (*Raman et al., 2010*). The patterned ORN responses we report here likely contribute substantially to the variety of this input, and thus to the complexity and high dimensionality of PN responses. In other species, how ORN response patterns are utilized downstream may depend on species-specific variations in connectivity between ORNs and the antennal lobe and its glomeruli.

Although our model is phenomenological, it was designed to simulate properties found in real neurons that can be traced to specific ion channels (*Komarov et al., 2018*). Also, earlier work

including biophysically realistic models has been shown to generate spiking behaviors comparable to the response motifs we identified (e.g., *Betkiewicz et al., 2020*; *Farkhooi et al., 2013*). Realistic biophysical models may point toward a more granular and specific mechanistic understanding of the responses we observed in ORNs.

In summary, we found that the responses of ORNs in the locust are organized into a discrete small set of spiking motifs, and that different odors can elicit different motifs from a given ORN. Multiple response motifs increase the dimensionality of the neural code for odors, and motif switching dramatically amplifies distinctions between similar chemical inputs, allowing for more successful classification. Each motif has its own adaptation profile, contributing to the encoding of temporal features of odor stimuli. These features provide benefits for olfactory tracking, such as determining distance to an odor source. Understanding and applying these processing features to stimulus classifying or tracking devices could lead to advancements in artificial intelligence and robotics.

## Materials and methods

### Electrophysiology

Recordings were made from 132 young adult locusts (*Schistocerca americana*) raised in a crowded colony. For experiments, locusts were secured with tape and wax into a Petri dish with one antenna exposed. An AgCl reference electrode was placed into the eye contralateral from the recording antenna. A saline-filled glass electrode (~10 μm, ~10 MΩ) was placed into the exposed antenna as a differential electrode. An identical electrode attached to a head stage (Axon instruments, X0.1LU) was placed at the base of a sensillum to acquire signals through an amplifier (Axon Instruments, Axoclamp 2b). The acquired signal was amplified ×3000 (Brownlee Precision, Model 440) and digitally sampled (National Instruments, USB-6212 and USB-6215 DAQ; Labview Software) at 10 kHz. Extracellular recordings from ORNs were made from different types of sensilla (sensilla chaetica and sensilla trichodea). Spike sorting was done offline with a whole wave algorithm (*Pouzat et al., 2002*) implemented in MATLAB (MathWorks). Sorted recordings were only included in the study if they were estimated to have <5% false positives and <5% false negatives during stimulation (*Hill et al., 2011*).

### Odor stimulation

The odor delivery system has been described previously (*Brown et al., 2005*). Briefly, a constant stream of dried and activated carbon-filtered air (0.9 l min⁻¹) was directed to the antenna through a plastic tube (6.5 mm inner diameter). A vacuum funnel (7 cm) was placed behind the animal to clear the odor space. Odorized air (0.5 L min⁻¹) was delivered by injecting air with a pneumatic pump (Reliable Pneumatic PicoPump, World Precision Instruments) into the head space of a 100 ml glass bottle containing odorant solutions diluted in mineral oil (JT Baker) to various concentrations, and then into the constant stream. The odorant chemicals (Sigma) used in this study are components of the locust diet, wheat grass: 1-octanol (OCT), 1-hexanol (HEX0.1, HEX1, HEX, HEX100; 0.1, 1, 10, and 100% by volume, respectively), and cyclohexanol (CYC). Pentyl acetate (PET), a naturally occurring chemical with an apple-like scent, was used as well.

Odorants other than 1-hexanol solutions were diluted to 10% by volume. The artificial plume stimulus was based on the burst length and burst return parameters derived from real odor plumes measured outdoors; see the 'Distance-based artificial plume generation' section (*Aldworth and Stopfer, 2015*; *Murlis et al., 2000*).

### Response motif clustering

We binned the spike-sorted responses of individual ORNs to 1 s odor pulses into 50 ms windows. Only responsive cells, those whose stimulus-evoked activity exceeded 2 standard deviations from the pre-stimulus mean, were included in the clustering analysis. A hierarchical cluster tree was created through unsupervised agglomerative hierarchical clustering ('linkage' function in MATLAB) using the Euclidean distance metric and the Ward agglomeration method. Responses were centered and normalized by the standard deviation before clustering.

We employed two tests to determine whether our clustering method separated responses patterns into statistically meaningful sets. Test 1 ensured that responses within clusters were more similar to each other than to random same-size subsets of all responses. Using a bootstrap approach, we chose

10,000 different random subsets of responses and, for each subset, calculated the mean distance between each response pair, yielding a distribution of mean distances. Each cluster was determined to be a statistically significant subset if the mean distance between response pairs within the cluster overlapped with less than 5% of the random distance distribution. Test 2 was used to determine whether each cluster was significantly distinct from all other clusters. We evaluated this by comparing intra- and inter-cluster distances with two-sample *t*-tests, using Bonferroni corrections to adjust for multiple comparisons. (*Figure 1—figure supplement 3*). It should be noted that this measure is nonsymmetric, changing with the cluster chosen for the intra-cluster distance comparison.

This analysis showed that response motifs formed significantly distinct clusters (passed Test 1) when we directed the algorithm to produce two, three, or four clusters. When five or more clusters were produced, the results failed Test 1 and/or Test 2. The choice of four clusters matched the impressions of experimenters viewing the data. Test 2, however, showed that inhibitory and offset clusters based on responses to 1 s stimuli were not always statistically distinct from one another, likely because floor effects limited the extent to which the inhibitory portions of these responses could vary, and because the 1 s odor stimulus elicited brief responses in which some onset and offset features overlapped. However, the same analysis applied to response motifs elicited by lengthier 4 s pulses of odorants always yielded statistically distinct inhibitory and offset clusters. Further, inhibitory and offset responses showed different adaptation profiles (see below), suggesting that they are driven, at least in part, by different mechanisms. Also, inhibitory and offset motifs would trigger different consequences downstream, with only the offset motif eliciting post-stimulus spiking in follower neurons. Thus, going forward, we included four distinct response motifs in our analysis.

## Quantification of motif switching

Motif switching rates were calculated by dividing the number of observed motif switches by the number of motif switching opportunities. The number of switching opportunities varied across subjects because some odor presentations did not elicit detectable responses from some ORNs.

## Adaptation analysis

Sensory adaptation was tracked by measuring the maximum absolute change in firing rate from baseline to a detection window (highlighted in gray, *Figure 3A*, *Figure 3—figure supplement 1*) that followed each odor pulse in the trial. The baseline was defined as the spike rate during the 2 s prior to the delivery of the first odor pulse in each trial. The maximum absolute change value following each odor pulse was then normalized within each trial by that trial's maximum firing rate. This was done to measure the proportion of change within each trial as an index of adaptation. Results were tested for significance by one-way repeated-measures ANOVAs.

## Model of ORN response

We designed ORN models based on a computationally efficient approach we previously proposed to simulate the neuronal dynamics of different types of neurons (*Bazhenov et al., 2008*; *Rulkov, 2002*; *Rulkov et al., 2004*). This phenomenological neuron model is described by a set of difference (map) equations and offers several numerical advantages: it avoids the common problem of selecting the proper integration scheme since the model is already written in the form needed for computer simulations; simulations with large time steps are stable and precise and tens to thousands of times faster than models based on differential equations (e.g., Hodgkin–Huxley models); model parameters can be adjusted to match experimental data (*Bazhenov et al., 2008*; *Rulkov and Bazhenov, 2008*; *Rulkov et al., 2004*). We previously used this approach to describe populations of locust Kenyon cells in *Assisi et al., 2020*; *Assisi et al., 2007*; *Kee et al., 2015*; *Sanda et al., 2016*. Here, briefly, difference equations, rather than ordinary differential equations, were used to generate a sequence of membrane potential samples at discrete time points with time step $h$ = 0.5 ms,

$$x_{n+1} = f_\alpha \left( x_n, y_n + \beta_n \right),$$
$$y_{n+1} = y_n - \mu \left( 1 + x_n - \sigma - \sigma_n \right),$$

(1)

where the variable $x_n$ given by the first equation modeled the fast dynamics of a neuron where each spike is formed by a single iteration with $x_n = 1$ due to the use of the discontinuous nonlinear function

$$f_\alpha(x_n, u) = \begin{cases} \frac{\alpha}{(1-x_n)} + u, & if\ x_n < -0.5, \\ 1, & if -0.5 \le x_n < 1, \\ -1, & if\ x_n \ge 1. \end{cases} \tag{2}$$

The second equation for variable $y_n$ controlled the slower ($0 < \mu \ll 1$) transient dynamics of the spiking activity and subthreshold oscillations. Intrinsic parameters α and σ define the baseline state and the regime of spiking in each neuron, determining whether the neuron produces tonic spiking or a burst of spikes. $\beta_n$ and $\sigma_n$ are input variables setting fast ($\beta_n$) and slow ($\sigma_n$) responsivity on the map model to external influences. $\beta^r$ and $\sigma^r$ are used to tune the shape of receptor response. For inhibitory cells, $\beta^r$ controls the level of rebound activity. It is the parameter driving the post-stimulus effect. For excitatory cells, it controls the fast response to the stimulus and the deceleration of spiking responses as follows:

$$\beta_n = \beta^\xi \xi_n + \beta^r I_n, \tag{3}$$

where coefficients $\beta^r$ and $\sigma^r$ in front of the stimulus $I_n$ were used to tune the shape of the receptor response for features such as speed of reaction in excitatory cells and the level of rebound activity in inhibitory cells. The noise $\xi_n$ in the ORN neurons was modeled as Ornstein–Uhlnebeck (OU) process:

$$\xi_{n+1} = q\xi_n - p\,w_n, \tag{4}$$

$p = d\sqrt{1-q^2}$, $h = 0.5$ ms, the iteration time step, $\tau_c$ is the correlation time in ms, and $d$ is the standard deviation of white Gaussian noise $w_n$. Parameters of the noise model were set to $\tau_c = 3$ ms, $d = 0.01$, $\beta^\xi = 0.1$, $\sigma^\xi = 1.0$. Distinct trials were generated using different random initialization for these noise processes.

The input current, $I_n$, was shaped using a first-order low-pass filter:

$$I_{n+1} = \gamma^s I_n + (1 - \gamma^s)\,a^s s_n, \tag{5}$$

where $s_n$ is the stimulus (odor concentration), the parameter $\gamma^s$ is the controlled relaxation time, and $a^s$ is the controlled responsivity strength and type (excitation or inhibition).

To model the different types of responses to an odor we observed in ORNs in vivo, we altered the parameters $\beta^r$ and $\gamma^s$, and $a^s$ defining response onset speed, and other equations to shape transient characteristics of input current $I_n$ with discrete-time filters based on the neuron motif. For a given motif, each of these parameters is fixed for all time and for all neurons of that motif.

| Response motif | $a^s$ | $\gamma^s$ | $\beta^r$ | $\sigma^r$ |
|---|---|---|---|---|
| Excitatory | 0.039 | 0.995 | 0.1 | 1 |
| Inhibited | –0.02 | 0.998 | 0.1 | 1 |
| Offset | –0.04 | 0.998 | 0.2 | 1 |

For example, the inhibited and offset motif are driven below baseline firing rate by stimulus due to the negative sign of the $a^s$ parameter. The delayed response motif features inhibition first, and, then excitation with a slow relaxation rate. To model the initial inhibition, we used a fast high-pass-filtered current and mixed it with the low-pass output to form a short negative pulse at the beginning.

$$I_{n+1}^d = \gamma^d I_n^d + a^d (s_n - s_{n-1}), \tag{6}$$

$$h_{n+1}^p = \gamma^h h_n^p + (s_n - s_{n-1}),$$

$$I_{n+1}^p = \gamma^p I_n^p + (1 - \gamma^p)\,a^p s_n h_{n+1}^p,$$

which are limited using the Heaviside function $H$ for $I_n^d$ and mixed with $I_n^p$

$$I_n^M = I_n^p - H\left(I_n^d\right) I_n^d \tag{7}$$

making input current for such receptors as

$$I_n = H\left(I_n^M + L\right) I_n^M,\tag{8}$$

where parameter $L$ in Heaviside limits the depth of inhibition at the beginning of response.

| Response motif | $a^p$ | $\gamma^p$ | $a^d$ | $\gamma^d$ | $\gamma^h$ | $L$ | $\beta^r$ | $\sigma^r$ |
|---|---|---|---|---|---|---|---|---|
| Delayed | 0.08 | 0.999 | 0.8 | 0.99 | 0.99985 | −0.04 | 0.01 | 1.0 |

## Model of odor selectivity

We used an angle-based method to model the interactions between ORNs and odors. Each model ORN was assigned a *selectivity vector*, $\mathbf{V_{ORN}}$ , and similarly each odor was assigned a *characteristic vector, $\mathbf{V_{odor}}$* . These vectors are defined in an arbitrary *n*-dimensional space. The efficacy or the odor–receptor interaction is given by *Raman et al., 2010*

$$R_i^A = \sigma\left(|V_{odor}| \cdot cos^p\left(\theta\right)\right)\tag{9}$$

where $\theta$ is the angle between $\mathbf{V_{ORN}}$ and $\mathbf{V_{odor}}$ in the *n*-dimensional space, $p$ is a parameter that defines the receptive field width of the receptor, and $\sigma\left(x\right)$ is the sigmoid response function defined as

$$\sigma\left(x\right) = \left[1 + exp\left(-a_1\left(x - a_2\right)\right)\right]^{-1}\tag{10}$$

with $a_1 = 15$, $a_2 = 0.3$. Consequently, odors whose characteristic vectors are close to a given ORN's selectivity vector will elicit a strong response in that ORN. Two odors that are close in this *n*-dimensional space are similar to each other and thus elicit similar responses across the population of ORNs.

Here, we defined the odor and receptor vectors in a 3D space, n = 3, with each coordinate $x_i \in [0, 1]$ (*Raman et al., 2010*). ORN response vectors were chosen randomly in the positive octant. For classification experiments, we defined a fixed initial odor vector (odor 1), we then created additional odor vectors at increasing angles of separation from the first odor. Odors were numbered by decreasing similarity to odor 1, for example, odor 2 is more similar to odor 1 than odor 9 to odor 1. All odor vectors were normalized.

## Principal component analysis

We created binned spike counts of model ORN responses using 50 ms bins. PCA was performed over these binned responses to reduce the 10,000-dimensional ORN space into two dimensions corresponding to the maximum explained variance for visualizing the odor trajectories. The trajectories were averaged over 10 trials for each odor. We used the *pca* function in MATLAB for this analysis.

## Classification of model odors

To classify odors, we first counted the spikes of model ORN responses into 50 ms bins. Binned response vectors were then used to train a SVM classifier to distinguish responses to different odors at each bin. We performed out-of-sample testing: the trial used for testing was chosen from 1 to 10, and the remaining 9 trials of each odor were used for training the SVM. Odor responses were classified at each 50 ms time bin and classification accuracy was averaged over the whole response. We constructed these SVMs using the *fitcecoc* function in MATLAB.

## Temporal filters using linear nonlinear models

Neural responses to temporally complex inputs have been modeled by combining a linear filter, $f\left(t\right)$, and a nonlinear threshold function, $g\left(x\right)$ (*Butts et al., 2007*; *Geffen et al., 2009*). Using this approach, the linear filter is convolved with the stimulus and then passed through a nonlinear thresholding function to generate the approximate firing rate, $r\left(t\right) = g\left[s\left(t\right) * f\left(t\right)\right]$. To derive the linear filter, we applied a standard deconvolution process: we deconvolved the signal, $s\left(t\right)$, from the true firing rates of ORNs following published techniques (*Butts et al., 2007*; *Geffen et al., 2009*). Briefly, trial-averaged cell responses were convolved with a Gaussian filter with width $\sigma = 50$ ms; deconvolution occurred by inverting the linear convolution matrix. To account for correlations occurring in naturalistic stimuli, we multiplied each filter by the inverse of the stimulus covariance matrix (*Sharpee, 2013*). Before

inverting the stimulus covariance matrix performed regularization using singular value decomposition and keeping components that account for 70% of the variance (*Sharpee et al., 2008*). The resulting filters were normalized by their $L_2$ norm and low-pass filtered with a stopband of 10 Hz. The nonlinear thresholding function, which connected the convolution product to the final estimate, was created by constructing a histogram of all potential outputs of the linear filtering. For each bin, $i$, in this histogram we averaged all linear products, forming $x_i$, and all true firing rates, forming $y_i$. This procedure created a series of inputs (averaged linear product values) and outputs (firing rate values) that defined the thresholding function, $y = g(x)$. For this analysis, we used 100 bins. The first 16 s of 40 s odor plumes were used for training and the next 16 s were used for testing. We then altered which 50% stretch of the data was used to train the model to derive a 95% confidence estimate of our L-NL prediction.

### Distance-based artificial plume generation

We generated artificial odor plumes based on experiments in which plumes of ionic tracers in outdoor settings were measured at various distances from the source (*Murlis et al., 2000*). The statistics of the plume burst lengths and inter-burst intervals in these published experiments closely matched gamma distributions with the appropriate length and shape parameters. We used the arithmetic and geometric mean statistics of the real plumes to generate gamma distributions for burst length and inter-burst intervals representing plumes measured 2.5 m, 5 m, 10 m, and 20 m from the source. This optimization was achieved by the following relationship for gamma distributions:

$$\log(k) - \psi(k) = \log(\mu) - \log(\gamma) \tag{11}$$

where $k$ is the shape parameter, $\psi$ is the polygamma function, μ is the arithmetic mean, and $\gamma$ is the geometric mean. This relationship was solved numerically using the MATLAB function *fsolve*. The length parameter was then given from the relationship $\theta = \mu/k$.

### Distance-based stimulus classification

We delivered 15 s trials of artificial plumes constructed to reflect different distances from the source (2.5 m, 5 m, 10 m, and 20 m) to simulated ORNs. The ORN responses were then used to classify the stimulus by distance to source. Spiking responses of each ORN were counted into 50 ms bins and were then concatenated across cells, resulting in *NCells * NBins* length feature vectors. SVM classifiers were then trained using these vectors. Equal numbers of trials were used for training or testing. Each SVM was trained only using cells generating a given set of response motifs. To compare results across different SVMs, we used the same number of cells (n = 192, the smallest number of responsive cells across all response motifs) regardless of the motifs used in the classifier. To ensure the specific selection of cells did not influence the results, we created 25 different SVMs based on different selected cells and randomly chosen trials for training or testing. As before, SVMs were trained using MATLABs *fitcecoc* function.

## Acknowledgements

Supported by ONR (N00014-16-1-2829), NIH (RF1MH117155 and R01NS109553), Intel Corp (CG42647565 FE2018), NSF (IIS-1724405), NIH (1R01DC020892), and NSF (EFRI BRAID 2223839) awarded to MB, a doctoral thesis grant from Obra Social La Caixa (ID 100010434 with code LCF/BQ/ES15/10360004) awarded to APM, and an intramural grant from the National Institute of Child Health and Human Development, National Institutes of Health (MS).

## Additional information

### Funding

| Funder | Grant reference number | Author |
|---|---|---|
| Office of Naval Research | N00014-16-1-2829 | Maxim Bazhenov |

| Funder | Grant reference number | Author |
|--------|------------------------|--------|
| National Institutes of Health | RF1MH117155 | Maxim Bazhenov |
| National Institutes of Health | R01NS109553 | Maxim Bazhenov |
| National Science Foundation | IIS-1724405 | Maxim Bazhenov |
| Obra Social La Caixa | ID 100010434 with code LCF/BQ/ES15/10360004 | Ana P Milan |
| Eunice Kennedy Shriver National Institute of Child Health and Human Development | Intramural | Mark A Stopfer |
| Intel Corp | CG42647565 FE2018 | Maxim Bazhenov |
| NIH | 1R01DC020892 | Maxim Bazhenov |
| NSF | EFRI BRAID 2223839 | Maxim Bazhenov |

The funders had no role in study design, data collection and interpretation, or the decision to submit the work for publication.

## Author contributions

Brian Kim, Seth Haney, Conceptualization, Resources, Data curation, Software, Formal analysis, Validation, Investigation, Visualization, Methodology, Writing - original draft, Writing - review and editing; Ana P Milan, Formal analysis, Investigation, Visualization, Methodology; Shruti Joshi, Data curation, Formal analysis, Validation, Visualization, Methodology, Writing - review and editing; Zane Aldworth, Conceptualization, Resources, Supervision, Validation, Methodology, Writing - review and editing; Nikolai Rulkov, Alexander T Kim, Data curation, Methodology; Maxim Bazhenov, Conceptualization, Resources, Software, Supervision, Funding acquisition, Project administration, Writing - review and editing; Mark A Stopfer, Conceptualization, Resources, Formal analysis, Supervision, Funding acquisition, Validation, Investigation, Writing - original draft, Project administration, Writing - review and editing

## Author ORCIDs

Zane Aldworth ![ORCID] http://orcid.org/0000-0002-0647-8465
Maxim Bazhenov ![ORCID] http://orcid.org/0000-0002-1936-0570
Mark A Stopfer ![ORCID] http://orcid.org/0000-0001-9200-1884

## Decision letter and Author response

Decision letter https://doi.org/10.7554/eLife.79152.sa1
Author response https://doi.org/10.7554/eLife.79152.sa2

# Additional files

## Supplementary files
• MDAR checklist

## Data availability

All data generated or analyzed during this study have been deposited at Open Science Framework and can be accessed here: Kim B and Haney S, Joshi S, Bazhenov M, Stopfer M (2022) Open Science Framework. Olfactory receptor neurons generate multiple response motifs, increasing coding space dimensionality https://osf.io/8bs72/.

The following previously published dataset was used:

| Author(s) | Year | Dataset title | Dataset URL | Database and Identifier |
|---|---|---|---|---|
| Kim B, Haney S, Joshi S, Bazhenov M, Stopfer M | 2022 | Olfactory receptor neurons generate multiple response motifs, increasing coding space dimensionality | https://osf.io/8bs72/ | Open Science Framework, 8bs72 |

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
