## [Editor Report]

This important work describes the temporal mechanisms of odor coding in the olfactory neurons of the locust. The supporting evidence is compelling and based on extensive experimental and computational analyses. This work will be of interest to sensory neuroscientists.

---

## [Decision Letter]

**Decision letter after peer review:**

Thank you for submitting your article "Olfactory receptor neurons generate multiple response motifs, increasing coding space dimensionality" for consideration by *eLife*. Your article has been reviewed by 3 peer reviewers, and the evaluation has been overseen by a Reviewing Editor and Piali Sengupta as the Senior Editor. The following individual involved in the review of your submission has agreed to reveal their identity: Martin Nawrot (Reviewer #3).

Essential revisions:

As you can see from the individual reviews, all reviewers (and myself) agree that this is a study of high potential importance with expertly executed experiments that result in an impressive and highly valuable dataset. However, we also all agree that some work is needed not only to expand the discussion but in particular to link the modelling part better to the experimental/data analysis one.

1) Presentation and level of detail: Please include detailed statistical descriptions throughout, details about the recording configuration including sensillum type, as well as detailed statistics about the interaction of motif switching and odour types. Do expand figure descriptions to ensure that all figure parts are properly explained.

2) Computational modelling: Locusts have a specific and unusual arrangement in that OSNs expressing the same receptor project to different glomeruli. Moreover, they have significantly more OSNs than e.g. *Drosophila*. Please assess how OSN number and the specific projection pattern affect the conclusions of the model to qualify its generality.

This is indeed a key issue – with the architecture of the locust's olfactory system so different from other insects (and lots of unknowns with respect to the molecular make-up), this not only requires extensive and detailed discussion and comparison but also possibly exploring larger parameter space in the model.

3) Computational modelling: Tighten the link between experimental approach and model, for example, by comparing the distribution of response latencies and peak rates. Assess the validity of the filters used and perform explicit cross-validation. Individual model parameters need to be explicitly described and it needs to be made clear how they impact the model predictions. Figure 5 seems to assess the consequence of motif-switching for a situation where responses to one odor are held constant. Importantly, the comparison to ORN activity in Figure 6 needs to be quantified.

4) Computational modelling: Generally, the model needs to be described much better, and accessible to non-experts. Equations need to be labelled, parameters described in detail, etc.

5) The set of odours used is quite limited. Ideally, a broader set of odours would be presented including odour mixtures and ethologically relevant "specialist" odours. Please at least discuss your findings and their applicability to other odour classes such as complex odour mixtures or ecologically particularly relevant odours where possibly a specialized circuitry could be expected.

6) As you can see from the detailed reviews, several important references and discussion points are missed. Please thoroughly go through those and amend accordingly.

*Reviewer #1 (Recommendations for the authors):*

This study would largely improve and be less specialized if the authors would broaden their discussion and provide further insight into their modeling approach. Furthermore, by expanding the odor set with ecologically relevant and rather diverse odors (such as PAN), it would be highly interesting to see whether the separation in these four response motifs would persist and whether odor identity coding would be more prominent. In addition, I have several specific suggestions that would improve readability and the value of the MS:

– The authors used two odors at different concentrations, while the other two odors were applied in just one concentration. Please add the concentration of the other two odors. It would be interesting to see the impact of odor concentrations on the dynamic of the OSN responses. Are the observed response motif concentration-dependent?

– The authors recorded mainly from trichoid sensilla, since those contain small numbers of OSNs which simplifies spike sorting. However, the authors mention that they also included recordings of other sensillum types. Please specify which types were measured and whether the response dynamics were sensillum-specific since those should express different types of ORs or IRs.

– In general, I am missing detailed info about the statistics used. The authors mentioned e.g. in Figure 3 that offset responses increased significantly during adaptation without providing any details. This increase is hardly visible in the corresponding figure.

– The descriptions of the figures in the figure legends are often insufficient and do not allow us to understand what is actually represented. For example, what is exactly shown in the two panels in Figure 3B? Does each line correspond to the same OSN responding to either hexanol or cyclohexanol? What is the meaning of the color code in Figure 3C? What are the shaded traces in Figure 4B? What is exactly represented in Figure 5C and what is the meaning of the color code? Please also increase the font size in Figure 3 to make it readable.

– The authors state that each OSN can exhibit different response motifs. How reliable/reproducible are the recorded responses? Do individual OSNs also switch between motifs for the same odor? Furthermore, it would be very informative if the motif switches would be analyzed with regard to odor specificity. Are the motif switches odor-dependent? Does for example an OSN that is tuned to hexanol always reveal an inhibition to cyclohexanol?

– As mentioned above, the computational modeling part is rather written for specialists and should be revised.

– Please cite and discuss the paper by Martelli and Fiala (*eLife*, 2019), which addresses adaptation mechanisms to odor pulses in OSNs in *Drosophila*.

*Reviewer #2 (Recommendations for the authors):*

Specific suggestions to address concerns are as follows:

1) There are 119 ORNs in the locust but simulations use a much greater number of receptors (10,000). Please justify this number or reduce it to match the experimental system.

2) The set of simulations in Figure 5 shows that motif switching improves odor classification. However, the comparison is made while holding responses to one odor constant while shifting responses to the second odor in different ways. This does not reflect the experimental situation where there is no motif switching for a given odorant.

3) The method that is used here to reconstruct neural filters is not appropriate for strongly correlated and natural odorant stimuli delivered experimentally. For a review of methods, please see https://pubmed.ncbi.nlm.nih.gov/23841838/.

4) It is imperative to add quantification for how accurately the model describes the ORN activity in Figure 6B.

5) Add error bars in Figure 6B for the estimated filters and gain functions.

*Reviewer #3 (Recommendations for the authors):*

Based on my opinion phrased in the public review above, I believe that this manuscript deserves publication with *eLife* if revised appropriately. That is, I have no doubt about the quality of experimental and theoretical methods and results. My concerns below refer to minor points in the description of the theoretical methods that may be improved, to a number of relevant but missing references including the observation of ORN response patterns in the fly, and to the Discussion that should be strengthened/deepened to further increase the impact of the present MS.

Specific review comments

1. ORN response motifs and temporal stimulus pattern responses in vivo and in silico

Galili et al. (2011) have shown clear offset responses (termed "post-odor responses") on ORNs in *Drosophila*. Martelli et al. (2013) have performed an extensive study on in vivo ORN responses and linear filter modeling of different stimulus-response patterns that partially fit the motif observations in the present manuscript. To my interpretation of their results, excitatory vs. delayed responses may also be odor concentration-dependent, different from the conclusion in the present supplementary figure. These studies should be cited and discussed; possibly there are additional references to that point that I am not aware of.

2. Discussion: ORN motifs in PN responses

PN responses have been shown to be highly complex in their odor-dependent temporal profile, in particular, they show inhibitory responses and delayed (late) responses (Krofczik et al., 2009), and off-responses (Galili et al., 2011). Individual PNs can respond rapidly to odor onset for one odor and exhibit a strong inhibition of other odors. This inhibition can be very fast abolishing PN response efficiently (Krofczik et al., 2009). The ORN motif switching shown here could provide an explanation for this observed behavior in PNs. Also, ORNs make direct connections to both, PNs and LNs (shown in detail in *Drosophila*) and this may further accentuate the expression of similar motifs in PNs odor code, e.g. if individual ORNs predominantly or exclusively target PNs and inhibitory LNs. Indicating the potential effect on PN coding in the Discussion will add to the impact of the present MS.

3. Discussion: mechanistic modeling of ORN adaptation in biophysical model neurons

The authors use a phenomenological linear filter model to describe the stimulus-response current. The Discussion does not indicate how realistic biophysical models for adaptive ORNs can at least capture the excitatory motif (as in classical stimulus adaptation reviewed in Benda 2021) and post-stimulus rebound effects. E.g the conductance-based spike frequency adaptation model (Farkhooi et al., 2013) has been shown to fit the excitatory response motif; this paper also showed that these mechanisms significantly increase response reliability across trials that cannot be captured by the phenomenological model presented here. This model of adaptation also explains the inhibitory post-stimulus effect after an excitatory response and the post-inhibition rebound as offset-response (Farkhooi et al., 2013, Betkiewicsz et al., 2020) that are also present in the response motifs where the avg. excitatory firing rate drops below baseline (Figure 1C, bottom and top) and in the inhibitory rebound after the offset of the long stimulus (Figure 1C, bottom, Figure 2A). The stimulus-response filters are designed to capture both these effects in the present MS (Figure 4B).

4. Discussion: Functional interpretation for odor sensing in a complex odor environment

The results are relevant for biologically realistic integrative models of the sensory pathway and sensory-motor transformations. The temporal dynamics of ORN responses are particularly relevant when simulating odor navigation behavior in flying or walking insects. The recent study by Rapp and Nawrot (2020) has shown that ORN adaptation (classical) translates to PN firing (which are modeled as non-adaptive neurons) and is important to reproduce temporally sparse coding in the mushroom body and is thus required for the active sampling of the statistics of odor encounters that can subserve navigation.

5. Discussion: Distance-related odor plume statistics

There have been several studies on the distance-dependent odor plume statistics and their mimicking in temporally patterned stimulation during physiological recordings, in particular in the moth (Jacob et al. 2017, Levakova et al. 2018).

6. Quantitative measures to inform models

For modeling purposes, it would be valuable if the authors can provide additional quantitative measures such as a distribution of response latencies and peak rates. Also, Figure 1C shows average response rates +-SEM for the 4 motifs in the overlay. A supplemental figure that shows trial-averaged average responses per unit in overlay separately for the four motives would allow for variability across neurons.

7. Method description of the theoretical model

– The methods section appears in front of the Results section. I believe it should appear after the Discussion according to *eLife* instructions.

– Please number equations.

– Eqn. following l.189: I understand u=(y_n+β_n) as in the first equation, correct? I was puzzled by the spike that "can" be produced if -0.5 < x_n < 1 and expected a stochastic process. The correct interpretation is that a spike is produced when threshold -0.5 is crossed I assume. x_n+1 is set to -1 after a spike, which is a reset. Can x_n become smaller than -1 due to input and noise or is it bound?

– Eqn. following l.207: Is γ^S < = 1 and > = 0? Is it a fixed parameter?

– The description of the effect of the model parameters β, γ, and α remains somewhat vague and very short in lines 212/213; it is not entirely transparent to me which parameter drives post-stimulus effects and whether γ is fixed or, likely, follows the stimulus step response.

– The authors show PID responses in Figure 6 for random stimulus trains. What do they look like for the long stimulus pulses? I would expect a low-pass type PID response (charging curve) and could this account for some of the low-pass filter properties in the model?

– Prediction with linear non-linear cascade model. It is not clear to me how cross-validation is performed here. The filter and transfer functions are estimated from the responses to the stochastic pulse presentation. How is the cross-validation done? Training on a set of trials (how large) and prediction on a different set? Even more convincing would be to train the model on the first half of the stimulus train and test on the second half. I understand that all animals were presented with the same stimulus. Can the authors train the model on individual neurons and predict the response of the pseudo population of non-simultaneously recorded cells? How well does the model work when the filter is estimated from single or repeated pulse presentations, does this easily transfer?

References:

Benda, J. (2021). Neural adaptation. Current Biology, 31(3), R110-R116.

Betkiewicz, R., Lindner, B., and Nawrot, M. P. (2020). Circuit and cellular mechanisms facilitate the transformation from dense to sparse coding in the insect olfactory system. Eneuro, 7(2).

Farkhooi, F., Froese, A., Muller, E., Menzel, R., and Nawrot, M. P. (2013). Cellular adaptation facilitates sparse and reliable coding in sensory pathways. PLoS computational biology, 9(10), e1003251.

Galili, D. S., Lüdke, A., Galizia, C. G., Szyszka, P., and Tanimoto, H. (2011). Olfactory trace conditioning in *Drosophila*. Journal of Neuroscience, 31(20), 7240-7248.

Jacob, V., Monsempès, C., Rospars, J. P., Masson, J. B., and Lucas, P. (2017). Olfactory coding in the turbulent realm. PLOS Computational Biology, 13(12), e1005870.

Krofczik, S., Menzel, R., and Nawrot, M. P. (2009). Rapid odor processing in the honeybee antennal lobe network. Frontiers in computational neuroscience, 2, 9.

Levakova, M., Kostal, L., Monsempès, C., Jacob, V., and Lucas, P. (2018). Moth olfactory receptor neurons adjust their encoding efficiency to temporal statistics of pheromone fluctuations. PLoS computational biology, 14(11), e1006586.

Martelli, C., Carlson, J. R., and Emonet, T. (2013). Intensity invariant dynamics and odor-specific latencies in olfactory receptor neuron response. Journal of Neuroscience, 33(15), 6285-6297.

Rapp, H., and Nawrot, M. P. (2020). A spiking neural program for sensorimotor control during foraging in flying insects. Proceedings of the National Academy of Sciences, 117(45), 28412-28421.

---

## [Author Response]

Essential revisions:As you can see from the individual reviews, all reviewers (and myself) agree that this is a study of high potential importance with expertly executed experiments that result in an impressive and highly valuable dataset. However, we also all agree that some work is needed not only to expand the discussion but in particular to link the modelling part better to the experimental/data analysis one.

Many thanks for this very positive and helpful assessment of our work! As we describe below, we’ve now made many revisions to improve our manuscript. Please note that line numbers given in this letter refer to the clean copy provided with this resubmission.

1) Presentation and level of detail: Please include detailed statistical descriptions throughout, details about the recording configuration including sensillum type, as well as detailed statistics about the interaction of motif switching and odour types. Do expand figure descriptions to ensure that all figure parts are properly explained.

We thank the editor and reviewers for their attention to our manuscript and for their helpful suggestions. As requested, we have now expanded these descriptions.

2) Computational modelling: Locusts have a specific and unusual arrangement in that OSNs expressing the same receptor project to different glomeruli. Moreover, they have significantly more OSNs than e.g. *Drosophila*. Please assess how OSN number and the specific projection pattern affect the conclusions of the model to qualify its generality.This is indeed a key issue – with the architecture of the locust's olfactory system so different from other insects (and lots of unknowns with respect to the molecular make-up), this not only requires extensive and detailed discussion and comparison but also possibly exploring larger parameter space in the model.

As the reviewer notes, it is clearly true that species-specific differences exist in olfactory circuitry downstream from OSNs. The responses of OSNs in other species, including *Drosophila*, have been shown to include more than one spiking pattern, as we discuss (lines 471-481). But, briefly, we feel strongly that a substantial expansion of our treatment of species-specific differences falls outside of the subject of our manuscript which addresses only responses of OSNs, not their followers.

For this project we did not record from follower neurons. Our computational analysis of OSN activity was designed to reveal the information content of OSN responses without making any assumptions about, or drawing upon any features of, follower neurons or their projection patterns. Our analysis, for example, does not in any sense include a model of glomeruli or the antennal lobe or any other aspect of the architecture of the locust brain. We agree that such questions are interesting. In fact, earlier work from our group (Raman et al., 2010) investigated downstream consequences of patterned activity in ORNs. In our revised manuscript we discuss these ideas in lines 337-346 and 480-490. But revising our model to include downstream olfactory structures, such as the antennal lobe and glomeruli, would essentially constitute a major new study, one likely to raise many new questions about the assumptions we would have to make. We do plan to begin such a study in the near future.

The reviewer is also correct that different species have different numbers of OSNs, and different numbers of types of olfactory receptor proteins (we acknowledged this in our manuscript (lines 57-61)). However, these differences also do not affect our conclusions. Only one calculation in our manuscript, odor classification accuracy, would be affected by numbers of OSNs and OSN types; increasing numbers of OSNs and OSN types would, from first principles, be expected to allow greater absolute classification success; we think readers will correctly assume this is so. However, our conclusions do not rely on absolute levels of classification accuracy. Rather, we focus on the *improvement in accuracy* provided by including multiple response motifs in the analysis. The fact of this improvement would not be affected by varying the numbers of OSNs or OSN types.

Because our conclusions do not depend to any extent upon the architecture of the locust's olfactory system or numbers of OSNs or OSN types, we would prefer to omit detailed and necessarily speculative discussions and analyses of these factors from our manuscript.

3) Computational modelling: Tighten the link between experimental approach and model, for example, by comparing the distribution of response latencies and peak rates. Assess the validity of the filters used and perform explicit cross-validation. Individual model parameters need to be explicitly described and it needs to be made clear how they impact the model predictions. Figure 5 seems to assess the consequence of motif-switching for a situation where responses to one odor are held constant. Importantly, the comparison to ORN activity in Figure 6 needs to be quantified.

These are excellent suggestions. We have now addressed all of them with new analyses, new figures, and extensive revisions to the text, as described in detail below.

4) Computational modelling: Generally, the model needs to be described much better, and accessible to non-experts. Equations need to be labelled, parameters described in detail, etc.

We agree and have revised descriptions of our model, throughout our manuscript, for greater accessibility.

5) The set of odours used is quite limited. Ideally, a broader set of odours would be presented including odour mixtures and ethologically relevant "specialist" odours. Please at least discuss your findings and their applicability to other odour classes such as complex odour mixtures or ecologically particularly relevant odours where possibly a specialized circuitry could be expected.

We thank the reviewers for raising this concern which often arises in olfactory studies. We have now expanded our discussion to address the applicability of our findings to complex odor mixtures and “specialist” odorants such as PAN in lines 401-405: “For these studies we used odorants known to be ecologically relevant to locusts, including several found in the head space of wheat grass. Future experiments with larger sets of odorants, including blends or locust pheromones like 4-vinylanisole (4VA) and phenylacetonitrile (PAN), may help clarify the logic of motif switching.”

We also note, as above, that our conclusions reflect only OSN responses, not possibly specialized circuitry that may follow.

6) As you can see from the detailed reviews, several important references and discussion points are missed. Please thoroughly go through those and amend accordingly.

We agree and have now extensively revised the text to add suggested references and discussion points.

Reviewer #1 (Recommendations for the authors):This study would largely improve and be less specialized if the authors would broaden their discussion and provide further insight into their modeling approach. Furthermore, by expanding the odor set with ecologically relevant and rather diverse odors (such as PAN), it would be highly interesting to see whether the separation in these four response motifs would persist and whether odor identity coding would be more prominent. In addition, I have several specific suggestions that would improve readability and the value of the MS:– The authors used two odors at different concentrations, while the other two odors were applied in just one concentration. Please add the concentration of the other two odors. It would be interesting to see the impact of odor concentrations on the dynamic of the OSN responses. Are the observed response motif concentration-dependent?

We thank the reviewer for these suggestions. We have extensively revised the manuscript to broaden our discussion and provide further insight into our modeling approach.

The reviewer’s suggestion to test additional odorants often arises in olfaction studies – there are always more odors to test and good reasons for testing them. We note, along with the other reviewers, that our dataset is already extensive, and we are reluctant to extend it further at present. However, to address the reviewer’s concerns we have now added the following text to our discussion in lines 401-405: “For these studies we used odorants known to be ecologically relevant to locusts, including several found in the head space of wheat grass. Future experiments with larger sets of odorants, including blends or locust pheromones like 4-vinylanisole (4VA) and phenylacetonitrile (PAN), may help clarify the logic of motif switching.”

We share the reviewer’s interest in ORN responses elicited by different concentrations of odors. Our revised manuscript provides an analysis of the impact of odor concentrations on the dynamic of the OSN responses in Figure 2 —figure supplement 1, and a description in the text in lines 122-126: “…we found very little switching across repeated trials of the same odor (p=0.096), and an intermediate amount of switching across concentrations of the same odor (p=0.25, see Figure 2 —figure supplement 1). Thus, motif switching appears to provide a mechanism to convey information about odorant identity and concentration that is robust to trial-to-trial variation.”

As this analysis shows, response motifs do vary with odor concentration, but only slightly, to a much lesser extent than variance due to changes in odor identity. Our discussion now expands on the idea that response motifs contribute to information about odor concentration in lines 386-390: “Even relatively subtle changes in odorant chemistry (hexanol vs cyclohexanol, for example), or changes in the concentration of an odor, could evoke changes in the response motifs of ORNs (immediate vs delayed spiking, for example), leading to dramatic differences in the timing of the odor response.”

– The authors recorded mainly from trichoid sensilla, since those contain small numbers of OSNs which simplifies spike sorting. However, the authors mention that they also included recordings of other sensillum types. Please specify which types were measured and whether the response dynamics were sensillum-specific since those should express different types of ORs or IRs.

We agree -- it is important to specify which sensillum types were measured and whether the response dynamics were sensillum-specific. Some of this information is already present in our manuscript, but to make this clear, our revision now states (in lines 92-94) that “All results other than those shown in Figure 1 —figure supplement 2 are based on recordings from trichoid sensilla.” Figure 1 —figure supplement 2 shows that “unsorted, population activity recorded from other types of sensilla yielded results consistent with the responses of sorted ORNs, including prominent onset and offset activity” (lines 90-92). Thus, as our manuscript now makes clear, the response dynamics we observed do not appear to be sensillum-specific.

– In general, I am missing detailed info about the statistics used. The authors mentioned e.g. in Figure 3 that offset responses increased significantly during adaptation without providing any details. This increase is hardly visible in the corresponding figure.

We agree. To address the reviewer’s concern, we have now revised throughout our text and figures to provide more information. Regarding the specific information requested for Figure 3, we found normalized offset responses increased from -1 to +1, a substantial and statistically significant change that can be seen in the histograms in panel A, and in the graph in panel B. To clarify this issue we revised the caption for Figure 3 to state: “Each ORN response motif has a distinct adaptation profile. (A) Responses of ORNs, grouped by motif, to odors pulsed at different inter-pulse intervals (IPIs). Peaks for different motifs and pulses were measured as maximum absolute change from baseline within detection windows (shaded areas). (B) Adaptation characteristics of excitatory, delayed, offset, and inhibitory response motif to 10 pulses for each IPI. Left: response motifs; Right: Normalized response change from baseline. *=statistically significant changes elicited by odor pulses delivered at 0.50s IPI. Excitatory motif responses significantly decreased; offset motif responses significantly increased; delayed motif responses modestly increased; and inhibitory motif responses did not change. See Table 1 for statistical tests” (lines 169-175).

– The descriptions of the figures in the figure legends are often insufficient and do not allow us to understand what is actually represented. For example, what is exactly shown in the two panels in Figure 3B? Does each line correspond to the same OSN responding to either hexanol or cyclohexanol? What is the meaning of the color code in Figure 3C? What are the shaded traces in Figure 4B? What is exactly represented in Figure 5C and what is the meaning of the color code? Please also increase the font size in Figure 3 to make it readable.

We thank the reviewer for this suggestion and have now revised our figure captions to include more information. We have also completely remade Figure 3 to make it more readable with an enlarged font. Briefly, the left panel in Figure 3B illustrates the response motif indicated by the label above it, and the right panel shows the normalized response in this motif’s change from baseline. The revised figure caption is given above. The shaded traces in Figure 4B indicate standard error of the mean; the caption now states this. In Figure 5C, darker shading indicates lower classification accuracy; the caption now states this.

– The authors state that each OSN can exhibit different response motifs. How reliable/reproducible are the recorded responses? Do individual OSNs also switch between motifs for the same odor?

Thank you – we have now revised our text to clarify these results. Briefly, the recorded responses are very reliable and reproducible, as can be seen in Figure 2. The revised text now states (lines 120-126): “Response motif switching was not rare, occurring with probabilities ranging from 0.31 to 0.66 (mean = 0.38) for different pairs of odors (see Figure 2B-D). By contrast, we found very little switching across repeated trials of the same odor (0.096), and an intermediate amount of switching across concentrations of the same odor (0.25, see Figure 2 —figure supplement 1). Thus, motif switching appears to provide a mechanism to convey information about odorant identity and concentration that is robust to trial-to-trial variation.”

Furthermore, it would be very informative if the motif switches would be analyzed with regard to odor specificity. Are the motif switches odor-dependent? Does for example an OSN that is tuned to hexanol always reveal an inhibition to cyclohexanol?

We thank the reviewer for these questions and have now revised the manuscript to make these results clearer and to provide appropriate context for them. Briefly, motif switches are to some extent odor specific. As shown in Figure 2C, an OSN responding with excitation to hexanol with a probability of 0.880 shows a 0.000 probability of responding to cyclohexanol with inhibition. Our results suggest an underlying logic to motif switching and odor specificity. For example, a chi-square analysis showed the distribution of odor motifs switches within odor pairs is not what one would expect by chance, indicating some underlying structure, but a rigorous analysis of this phenomenon would require a much larger dataset with many more odorants. We plan to address this phenomenon in a future study. We have now expanded our discussion to address these important points.

Our revised discussion now states: “Our experiments did not allow us to test whether all ORNs are capable of motif switching, or what, if any, organizational principle determines when a given ORN will switch motifs. But because some odor pairs were more likely than others to elicit motif switching, we speculate that motif switching may follow a logic that evolved to help discriminate ecologically relevant but chemically similar odors” (lines 397-401).

– As mentioned above, the computational modeling part is rather written for specialists and should be revised.

We thank the reviewer for this suggestion. We would like our work to be accessible to many readers and have revised the computational modeling part in several places for clarity.

– Please cite and discuss the paper by Martelli and Fiala (eLife, 2019), which addresses adaptation mechanisms to odor pulses in OSNs in *Drosophila*.

Thank you. We now cite and describe this paper (see line 401 and lines 438-401): “Our analysis did not investigate mechanisms underlying the different forms of adaptation we observed; a recent study of ORNs in *Drosophila* has revealed calcium dynamics play a role in adaption (Martelli et al., 2019).”

Reviewer #2 (Recommendations for the authors):Specific suggestions to address concerns are as follows:1) There are 119 ORNs in the locust but simulations use a much greater number of receptors (10,000). Please justify this number or reduce it to match the experimental system.

The exact number of ORNs in the locust is not known, but estimates range from 45,000 to 113,000 per antenna (Leitch and Laurent 1996; Perez-Orive et al. 2002; Galizia and Sachse 2010). We believe our choice to model 10,000 ORNs is a reasonable compromise between the ideal size (which would be true number of ORNs in locust) and practical limitations needed to achieve computational efficiency.

The referee may be thinking about numbers of types of ORs rather than numbers of ORNs. Our simulations did not explicitly organize ORs into types. However, we are confident that doing so would not affect our conclusions. Our model is phenomenological, designed to simulate the four spiking motifs we observed in vivo*.* This model does not rely to any extend on the receptor proteins and cellular mechanisms determined by OR types and that generate these spiking motifs in vivo. Several figures in our manuscript (Figures 4, Figure 4 —figure supplement 1, Figure 6) demonstrate our model does a good job of simulating these four motifs. It is only these spiking patterns, not the cellular mechanisms underlying them, that matter to our conclusions. Overall, we believe our approach is appropriate for our questions and conclusions.

2) The set of simulations in Figure 5 shows that motif switching improves odor classification. However, the comparison is made while holding responses to one odor constant while shifting responses to the second odor in different ways. This does not reflect the experimental situation where there is no motif switching for a given odorant.

We regret that our description of this analysis was unclear, as mentioned above (public review, comment #2). As described in detail above, we have now revised our text to clarify our goals and procedures (lines 191-195, 206-209).

3) The method that is used here to reconstruct neural filters is not appropriate for strongly correlated and natural odorant stimuli delivered experimentally. For a review of methods, please see https://pubmed.ncbi.nlm.nih.gov/23841838/.

We agree and thank the reviewer for raising this important point and recommending an appropriate analysis.

To account for natural correlations present in the stimuli we used in experiments, we have now completely redone our analysis, using the method suggested by the reviewer. Briefly, filters calculated through deconvolution were multiplied by the inverse stimulus covariance matrix as described in the paper cited (Sharpee, 2013). The results of this new analysis, shown in revised Figure 6, are consistent with results in the previous version of our manuscript.

This new analysis is described in the methods section of the revised manuscript, as follows: “To account for correlations occurring in naturalistic stimuli, we multiplied each filter by the inverse of the stimulus covariance matrix (Sharpee, 2013). Before inverting the stimulus covariance matrix performed regularization using singular value decomposition and keeping components that account for 70% of the variance (Sharpee et al., 2008). The resulting filters were normalized by their L_2_ norm and low-pass filtered with a stopband of 10Hz” (lines 713-718).

4) It is imperative to add quantification for how accurately the model describes the ORN activity in Figure 6B.

We agree and thank the reviewer for raising this important point. To qualitatively compare ORN activity in vivo and in the model we have now prepared new supplementary figure; Figure 4 —figure supplement 1. As shown, results obtained in vivo and in the model are highly correlated, indicating the model accurately describes ORN activity. We now state this directly in the text, as follows: “As desired, our model provided an accurate simulation of ORN responses observed in vivo (Figure 4 —figure supplement 1 provides a quantitative comparison of response latencies and peaks in vivo and in the model)” (lines 164-167).

5) Add error bars in Figure 6B for the estimated filters and gain functions.

Thank you. We have now revised the figure to include shading that indicates 95% confidence intervals.

Reviewer #3 (Recommendations for the authors):Based on my opinion phrased in the public review above, I believe that this manuscript deserves publication with eLife if revised appropriately. That is, I have no doubt about the quality of experimental and theoretical methods and results. My concerns below refer to minor points in the description of the theoretical methods that may be improved, to a number of relevant but missing references including the observation of ORN response patterns in the fly, and to the Discussion that should be strengthened/deepened to further increase the impact of the present MS.Specific review comments1. ORN response motifs and temporal stimulus pattern responses in vivo and in silicoGalili et al. (2011) have shown clear offset responses (termed "post-odor responses") on ORNs in *Drosophila*. Martelli et al. (2013) have performed an extensive study on in vivo ORN responses and linear filter modeling of different stimulus-response patterns that partially fit the motif observations in the present manuscript. To my interpretation of their results, excitatory vs. delayed responses may also be odor concentration-dependent, different from the conclusion in the present supplementary figure. These studies should be cited and discussed; possibly there are additional references to that point that I am not aware of.

We thank the reviewer for raising this interesting point. We now cite and discuss these studies. Briefly, we think our results are readily reconciled with those of Galili et al. and Martelli et al. by considering intrinsic odor dynamics. These earlier groups showed that odorants can have intrinsic dynamic properties and can be “fast” or “slow.” Our PID recordings showed none of the odors we used were “slow” – that is, none elicited PID signals that increased gradually throughout a stimulus presentation. Rather, all our odors elicited “fast” PID responses that approximated square pulses at all tested concentrations (please see the example below in response to point #7). Our observation rules out the possibility that differences in intrinsic odor dynamics could explain the origin of the response motifs we observed, and likely also explains why we did not observe the concentration-driven changes in ORN responses reported earlier.

2. Discussion: ORN motifs in PN responsesPN responses have been shown to be highly complex in their odor-dependent temporal profile, in particular, they show inhibitory responses and delayed (late) responses (Krofczik et al., 2009), and off-responses (Galili et al., 2011). Individual PNs can respond rapidly to odor onset for one odor and exhibit a strong inhibition of other odors. This inhibition can be very fast abolishing PN response efficiently (Krofczik et al., 2009). The ORN motif switching shown here could provide an explanation for this observed behavior in PNs. Also, ORNs make direct connections to both, PNs and LNs (shown in detail in *Drosophila*) and this may further accentuate the expression of similar motifs in PNs odor code, e.g. if individual ORNs predominantly or exclusively target PNs and inhibitory LNs. Indicating the potential effect on PN coding in the Discussion will add to the impact of the present MS.

Thank you -- we agree that it is important to consider the consequences of ORN motif switching on their downstream followers such as PNs. Our manuscript already included some of this discussion. In lines 337-344 we described how the responses of several layers of follower neurons are shaped by heterogeneities in the temporal structures of odor-elicited ORN responses, and cited our earlier work Raman et al., 2010 and Gupta and Stopfer, 2014, in support. Also, in Discussion (lines 480-483) we wrote: “ORNs provide the olfactory system’s first of several stages of signal processing. In insects, ORNs pass information to the antennal lobe, where spiking patterns originating in the periphery drive further processing in networks of local and projection neurons, leading to richer, higher dimensional olfactory representations.” As noted above, we predict interactions of ORN response motifs with intrinsic odor dynamics would likely increase ORN response dimensionality, and that greater response heterogeneity would further increase the dimensionality of PN responses.

To more directly address this point we have added the following to our discussion: “Our study did not address how multiple and switching response motifs in ORNs affect the responses of downstream neurons such as LNs and PNs. Our earlier work established that antennal lobe circuitry generates high-dimensional, complex temporally structured responses, but only when it is driven by input with heterogeneous timing from ORNs (Raman et al., 2010). The patterned ORN responses we report here likely contribute substantially to the variety of this input, and thus to the complexity and high dimensionality of PN responses” (lines 484-490).

3. Discussion: mechanistic modeling of ORN adaptation in biophysical model neuronsThe authors use a phenomenological linear filter model to describe the stimulus-response current. The Discussion does not indicate how realistic biophysical models for adaptive ORNs can at least capture the excitatory motif (as in classical stimulus adaptation reviewed in Benda 2021) and post-stimulus rebound effects. E.g the conductance-based spike frequency adaptation model (Farkhooi et al., 2013) has been shown to fit the excitatory response motif; this paper also showed that these mechanisms significantly increase response reliability across trials that cannot be captured by the phenomenological model presented here. This model of adaptation also explains the inhibitory post-stimulus effect after an excitatory response and the post-inhibition rebound as offset-response (Farkhooi et al., 2013, Betkiewicsz et al., 2020) that are also present in the response motifs where the avg. excitatory firing rate drops below baseline (Figure 1C, bottom and top) and in the inhibitory rebound after the offset of the long stimulus (Figure 1C, bottom, Figure 2A). The stimulus-response filters are designed to capture both these effects in the present MS (Figure 4B).

This is an excellent point and we thank the reviewer for raising it. Although our model is basically descriptive and phenomenological, earlier work including biophysically realistic models can generate spiking behaviors comparable to the response motifs we identified. More realistic models may point toward a more granular and specific mechanistic understanding of the phenomena we describe.

It is also worth mentioning that, while our model is phenomenological, it was designed to simulate properties found in real neurons that can be traced to specific ion channels. In essence, the first equation (x variable) describes fast spiking dynamics and thus represent effects of fast Na^+^, K^+^ currents (I_Na_, I_K_). The second equation describes slow membrane adaptation (variable y) that captures responses commonly represented by ca^2+^ dependent potassium currents (I_K(Ca)_), as well as rebound burst effects associated with low-threshold ca^2+^ currents (I_T_) found in some classes of neurons. Thus, in principle, the model suggests that a minimal set of currents needed to reproduce the excitatory and offset ORN responses would include I_Na_, I_K_, I_K(Ca)_, I_T_. Some additional voltage-gated K^+^ currents may be needed to explain delayed responses. These and other properties of the model are discussed in more detail here: Maxim Komarov, Giri Krishnan, Sylvain Chauvette, Nikolai Rulkov, Igor Timofeev and Maxim Bazhenov (2018) New class of reduced computationally efficient neuronal models for large-scale simulations of brain dynamics, *Journal of Computational Neuroscience*, 44:1–24.

To address the reviewer’s point we have expanded our discussion as follows: “Although our model is phenomenological, it was designed to simulate properties found in real neurons that can be traced to specific ion channels (Komarov et al., 2018). Also, earlier work including biophysically realistic models have been shown to generate spiking behaviors comparable to the response motifs we identified (e.g., Farkhooi et al., 2013; Betkiewicsz et al., 2020). Realistic biophysical models may point toward a more granular and specific mechanistic understanding of the responses we observed in ORNs” (lines 494-500).

4. Discussion: Functional interpretation for odor sensing in a complex odor environmentThe results are relevant for biologically realistic integrative models of the sensory pathway and sensory-motor transformations. The temporal dynamics of ORN responses are particularly relevant when simulating odor navigation behavior in flying or walking insects. The recent study by Rapp and Nawrot (2020) has shown that ORN adaptation (classical) translates to PN firing (which are modeled as non-adaptive neurons) and is important to reproduce temporally sparse coding in the mushroom body and is thus required for the active sampling of the statistics of odor encounters that can subserve navigation.

This paper is indeed relevant to our work and we now discuss it in our manuscript, as follows: “A recent modeling study found that adaptation of ORN responses contributes substantially to the sparsening of responses downstream, and to the sampling of the statistics of odor encounters that could aid navigation to food sources (Rapp and Nawrot 2020)” (lines 462-465).

5. Discussion: Distance-related odor plume statisticsThere have been several studies on the distance-dependent odor plume statistics and their mimicking in temporally patterned stimulation during physiological recordings, in particular in the moth (Jacob et al. 2017, Levakova et al. 2018).

We thank the reviewer for pointing us to these earlier, relevant studies. We now mention and cite these additional papers in two places as follows: “Earlier work performed in vivo and with computational models have investigated ways insect olfactory systems encode distance-dependent plume statistics (e.g., Jacob et al., 2017; Levakova et al. 2018)” (lines 460-462); and “A common theme of these findings is that olfactory systems include mechanisms to extract information needed for navigation” (lines 465-466).

6. Quantitative measures to inform modelsFor modeling purposes, it would be valuable if the authors can provide additional quantitative measures such as a distribution of response latencies and peak rates. Also, Figure 1C shows average response rates +-SEM for the 4 motifs in the overlay. A supplemental figure that shows trial-averaged average responses per unit in overlay separately for the four motives would allow for variability across neurons.

We agree and have now added two supplementary figures to provide this information. Figure 4 —figure supplement 1, as requested, compares the distributions of response latencies and peak rates observed for each motif in vivo and in our model. The caption: “Distribution of peak firing rates and response latencies for different motifs in model and in vivo experiments for a stimulus time period of 1000ms. (A) The peak responses were calculated as the maximum firing rate for the Excitatory, Delayed and Offset motifs, and as the minimum firing rate for the Inhibitory motif. (B) The response latencies were calculated as the time required to reach the peak response.”

A second new supplementary figure, Figure 1 —figure supplement 4, as requested, shows trial-averaged average responses per unit in overlay separately for the four motifs to illustrate variability across neurons. The caption: “1 second odor-olfactory receptor neuron responses. A total of 198 O-ORN responses are plotted in thin grey lines. The average response for each motif is overlaid in a thicker color line. Black bar denotes stimulus delivery.”

7. Method description of the theoretical model– The methods section appears in front of the Results section. I believe it should appear after the Discussion according to eLife instructions.

Fixed – thank you!

– Please number equations.

Done.

– Eqn. following l.189: I understand u=(y_n+β_n) as in the first equation, correct? I was puzzled by the spike that "can" be produced if -0.5 < x_n < 1 and expected a stochastic process. The correct interpretation is that a spike is produced when threshold -0.5 is crossed I assume. x_n+1 is set to -1 after a spike, which is a reset. Can x_n become smaller than -1 due to input and noise or is it bound?

Yes, u=yn+βn is the first equation. The interpretation that a spike is produced when −0.5<x< 1 is correct – this is not a stochastic process. xn is not bound by -1 on the lower side as it can become smaller due to noise and other inputs. This equation is meant to ensure correct spike generation in the discrete-time model. Setting xn+1 to -1 ends the spike event but does not reset the system -- that is achieved by two-dimensional dynamics of the model determined by variables yn and βn. More information about the dynamics of the model can be found in cited references. To clarify this in the revised manuscript we no longer use u in the equation for nonlinear functions, and we have revised our explanation as follows, in lines 616-621:

“Here, briefly, difference equations, rather than ordinary differential equations, were used to generate a sequence of membrane potential samples at discrete time points with time step h= 0.5ms,

 xn+1=fα(xn,yn+βn),          yn+1=yn−μ(1+xn−σ−σn), 

where the variable xn given by the first equation modeled the fast dynamics of a neuron where each spike is formed by a single iteration with xn=1 due to the use of the discontinuous nonlinear function fα(xn,u)={α(1−xn)+u,if xn<−0.5,1,if−0.5≤ xn<1,−1,if xn≥1.

– Eqn. following l.207: Is γ^S < = 1 and > = 0? Is it a fixed parameter?

Yes, γs is a fixed parameter between 0 and 1 to define the model’s relaxation rate. It is fixed for a particular motif and does not change with the input. The actual values of this parameter are given in the table beginning on line 650. We now mention this in the revised text as follows: “For a given motif, each of these parameters is fixed for all time and for all neurons of that motif” (lines 646-647).

– The description of the effect of the model parameters β, γ, and α remains somewhat vague and very short in lines 212/213; it is not entirely transparent to me which parameter drives post-stimulus effects and whether γ is fixed or, likely, follows the stimulus step response.

We agree it is important to clarify these points, and now include, in many small changes throughout the Methods, the following information:

– Parameters α and σ define the baseline state and the regime of spiking in each neuron, determining whether the neuron produces tonic spiking or a burst of spikes.

– βn and σn are input variables setting fast (βn) and slow (σn) responsivity on the map model to external influences. βr and σr are used to tune the shape of receptor response. For inhibitory cells, βr controls the level of rebound activity. It is the parameter driving the post-stimulus effect. For excitatory cells it controls the fast response to the stimulus and the deceleration of spiking responses.

– γs is a fixed parameter for a particular motif and controls the relaxation time of the response. It does not follow the stimulus step response.

– as is also a fixed parameter for a particular motif and controls the responsivity strength and the type (excitation or inhibition).

We have added to the text to explain which parameters drive peri-stimulus effects in inhibitory and offset motifs. For example, both motifs are driven below baseline firing rates by setting the as parameter negative.

– The authors show PID responses in Figure 6 for random stimulus trains. What do they look like for the long stimulus pulses? I would expect a low-pass type PID response (charging curve) and could this account for some of the low-pass filter properties in the model?

As we describe above, the PID responses we recorded to our odors had fast temporal profiles with rapidly rising and falling edges, as responses to long stimuli in Author response image 1 illustrate (4 sec stimulus pulse, green: individual traces; black: average). Because we observed throughout our dataset that every odor could elicit each of the four response motifs, we can conclude that the temporal properties of the odorants do not account for the model’s filter properties.

**Author response image 1. sa2fig1:** 

– Prediction with linear non-linear cascade model. It is not clear to me how cross-validation is performed here. The filter and transfer functions are estimated from the responses to the stochastic pulse presentation. How is the cross-validation done? Training on a set of trials (how large) and prediction on a different set? Even more convincing would be to train the model on the first half of the stimulus train and test on the second half. I understand that all animals were presented with the same stimulus. Can the authors train the model on individual neurons and predict the response of the pseudo population of non-simultaneously recorded cells? How well does the model work when the filter is estimated from single or repeated pulse presentations, does this easily transfer?

Cross-validation was performed just as suggested by the reviewer: we used the first 16sec of 40sec odor plumes for training and the next 16sec for testing. We now describe this in the text (lines 723-725).

We don’t have any evidence for a mechanism to correlate firing among ORNs in different sensilla, but we note that in other sensory systems (such as vertebrate retina) correlated firing between afferent neurons accounts for a small but significant amount of stimulus encoding (Latham and Nirenberg, 2005; Paninski et al., 2008). Because of this we treat our model, one trained on individual neurons recorded separately, as providing a lower bound estimate of the stimulus information contained in the population response.